# The thermal response of small and shallow lakes to climate change: new insights from 3D hindcast modelling.

Francesco Piccioni[1], Céline Casenave[2], Bruno Jacques Lemaire[1,3], Patrick Le Moigne[4], Philippe Dubois[1], and Brigitte Vinçon-Leite[1]

[1]LEESU, Ecole des Ponts ParisTech, Univ Paris Est Créteil, Marne-la-Vallée, France
[2]MISTEA, Université Montpellier, INRAE, Institut Agro, Montpellier, France
[3]AgroParisTech, Paris, France
[4]CNRM, Université de Toulouse, Météo-France, CNRS, Toulouse, France

**Correspondence:** Francesco Piccioni (francesco.piccioni@enpc.fr)

**Abstract.** Small and shallow lakes represent the majority of inland freshwater bodies. However, the effects of climate change on such ecosystems have rarely been quantitatively addressed. We propose a methodology to evaluate the thermal response of small and shallow lakes to long-term changes in the meteorological conditions, through model simulations. To do so, a 3D thermal-hydrodynamic model is forced with meteorological data and used to hindcast the evolution of an urban lake in the Paris region between 1960 and 2017. Its thermal response is assessed through a series of indices describing its thermal regime in terms of water temperature, thermal stratification and potential cyanobacteria production. These indices and the meteorological forcing are first analysed over time to test the presence of long-term monotonic trends. 3D simulations are then exploited to highlight the presence of spatial heterogeneity. The analyses show that climate change has strongly impacted the thermal regime of the study site. Its response is highly correlated with three meteorological variables: air temperature, solar radiation and wind speed. Mean annual water temperature shows a considerable warming trend of 0.6°C.dec$^{-1}$, accompanied by longer stratification and by an increase of thermal energy favourable to cyanobacteria proliferation. The strengthening of thermal conditions favourable for cyanobacteria is particularly strong during spring and summer, while stratification increases especially during spring and autumn. The 3D analysis allows to detect a sharp separation between deeper and shallower portions of the basin in terms of stratification dynamics and potential cyanobacteria production. This leads to the development over time of certain areas in the study site that are particularly favourable to cyanobacteria growth and bloom initiation.

## 1 Introduction

Lakes and reservoirs represent 3.7% of the Earth's non-glaciated continental area (Verpoorter et al., 2014), and often act as "sentinels" of climate change (Adrian R. et al., 2009). They have experienced considerable warming along the past decades (O'Reilly et al., 2015; Schmid et al., 2014; Schneider and Hook, 2010; Piccolroaz et al., 2020), sometimes even accelerated in respect to the surrounding areas (Schneider et al., 2009). Climate change is expected to further deteriorate the ecological status of a number of lakes worldwide that already suffer from eutrophication. In particular, changes in water temperature and in the patterns of thermal stratification could have a strong influence on the development of harmful algal blooms. Warmer water

temperatures might favor the dominance of certain phytoplankton species, such as cyanobacteria, whose increasing occurrence is a great concern in the management of water resources (Paerl and Huisman, 2008; Paerl and Paul, 2012; Wagner and Erickson, 2017). Furthermore, changes in the stratification and mixing regime could alter nutrients and light availability, sedimentation rates, and enhance the risk of hypolimnetic oxygen depletion (Song et al., 2013; Wilhelm and Adrian, 2008; Jankowski et al., 2006; Winder and Sommer, 2012).

The global areal extent of lakes and impoundments is dominated by millions of water bodies smaller than 1 km$^2$ (Downing et al., 2006). These small lakes have therefore to be taken into account in large-scale climate change analysis (Downing et al., 2006) and elemental budgets, such as the carbon budget (Mendonça et al., 2017). With the advance of urbanization, the presence of aquatic environments has become a key feature for the improvement of life quality in the urban landscape (Frumkin et al., 2017; van den Bosch and Sang, 2017). Often small and shallow (i.e. surface < 1 km$^2$, with light potentially penetrating to the bottom (Meerhoff and Jeppesen, 2009)), urban lakes grant valuable ecosystem services and contribute to the preservation of biodiversity (Frumkin et al., 2017; Hill et al., 2017; Hassall, 2014; Higgins et al., 2019). They often are prone to ecological deterioration and harmful algal blooms (Biggs et al., 2016; Wilkinson et al., 2020). For these reasons, in recent years small polymictic lakes are gaining greater attention in scientific studies. However, to our best knowledge, only a few studies can be found on the effect of climate change on such small and shallow water bodies (Biggs et al., 2016; Tan et al., 2018; Shatwell et al., 2019), whereas deeper monomictic or dimictic water bodies have received more attention. This lack of scientific studies is mirrored in a general lack of long-term *in situ* data, making it impossible to directly analyze how these environments respond to climate change solely through observations. Conversely, long-term meteorological data are available for most regions of the globe (e.g. global or regional reanalysis), as a result of a network of systematic observations that developed consistently since the beginning of the 20th century. These meteorological data can be used as external forcings in models, whose results will enable to fill in gaps in sporadic series of observation or to gain knowledge outside of the observation period (Magee and Wu, 2017; Vinçon-Leite et al., 2014; Kerimoglu and Rinke, 2013; Hadley et al., 2014). For a number of small and shallow lakes, this would enable to compensate the lack of observation data, making it possible to evaluate their response to climate change.

Hydrodynamic models have been vastly used to simulate lake and reservoir thermal dynamics over both short and long periods in order to test changes in systems subject to given meteorological and border conditions, often through one-dimensional simulations. However, most water bodies present a complex morphology, whose effects on the hydrodynamics can only be taken into account by three-dimensional models. This is crucial to study the presence of local patterns and spatial heterogeneity, and to reconstruct the lake dynamics not only in time but in space as well. In particular, the hydrodynamics and thermal regime of small and shallow lakes is complex and strongly influenced by meteorological conditions. They are usually polymictic and cannot be simply considered as completely mixed reactors (McEnroe et al., 2013). In fact, they alternate periods of complete mixing to periods of stable thermal stratification that, depending on the local meteorological conditions, can last up to a few weeks (Soulignac et al., 2017).

In this paper, we propose to use 3D thermo-hydrodynamic models to evaluate the thermal response of small and shallow lakes to climate change. The objective is to characterize the evolution of their thermal regime in relation to stratification dynamics and potential primary production, focusing in particular on cyanobacteria. To do so, the thermal dynamics of a small urban lake

was reconstructed along the past six decades (namely from 1960 to 2017) through a three-dimensional model. In addition to temperature values, a series of indices that are well-adapted to the specificities of the thermal regime of small and shallow lakes has been proposed to characterize the stratification dynamics and phytoplankton growth. The presence of long-term trends and the evolution of spatial heterogeneity of these indices were assessed. Although the proposed methodology was here applied to a study site located in the Paris region, it is generic and could be applied to other similar sites.

## 2 Materials and methods

### 2.1 Study site and in situ measurements

Lake Champs-sur-Marne is a sand-pit lake located in the East side of the Great Paris region, next to the Marne River. It is a small and shallow water body with a surface of 0.12 km$^2$, mean depth of 2.5 m and maximum depth of 3.5 m. As shown in Fig. 1-b, the southern part of the lake is the deepest one, while depth decreases under 2 m around the island and in the northern part of the lake. Lake Champs-sur-Marne has no inflow or outflow and is fed primarily by groundwater and occasionally by rainfall runoff. Its water level varies weakly during the year, with monthly oscillations lower than 0.2 m on average.

The lake was originated in the 1940s by excavation and represents now a valuable and demanded recreational area. However, it suffers from strong eutrophic conditions and experiences severe harmful algal blooms, especially between late spring and early autumn. In particular, cyanobacteria such as *Microcystis* and *Aphanizomenon*, capable to produce toxins, often proliferate and become the dominant species in the lake. This leads regularly to bathing bans and to restrictions in the access to the lake. For these reasons, the lake is subject to a periodic monitoring. A high-frequency (every 10 minutes) *in situ* measuring system was installed at two different locations (A and B) during the spring 2015. Each measuring site is equipped with sensors at three depths: below the surface at 0.5 m depth, in the middle of the water column at 1.5 m and above the sediment at 2.5 m (Tran Khac et al., 2018). Water temperature is recorded at the surface and bottom layers with a precision of 0.02 °C and a resolution of 0.05 °C through the thermal sensor SP2T10 (nke INSTRUMENT®), and through the MPx multi-parameter sensor (nke INSTRUMENT®) at the middle of the water column, with a precision and a resolution of 0.05°C (see Fig. 1-a). High-frequency water temperature observations are used here for the calibration and validation of the hydrodynamic model.

Lake Champs-sur-Marne is polymictic and its thermal behavior is strongly influenced by meteorological conditions. Between March and November periods of thermal stratification alternate with mixing and overturn of the water column. Depending on meteorological conditions, thermal stratification might form during the day and break up at night as well as last up to two or three consecutive weeks.

### 2.2 The model

#### 2.2.1 Presentation of Delft3D-FLOW

The hydrodynamics of the study site were simulated with the FLOW module of the Delft3D modelling suite (Deltares, 2014). Delft3D-FLOW is a well known hydrodynamic model that has been applied in various contexts, from estuaries to rivers, lakes

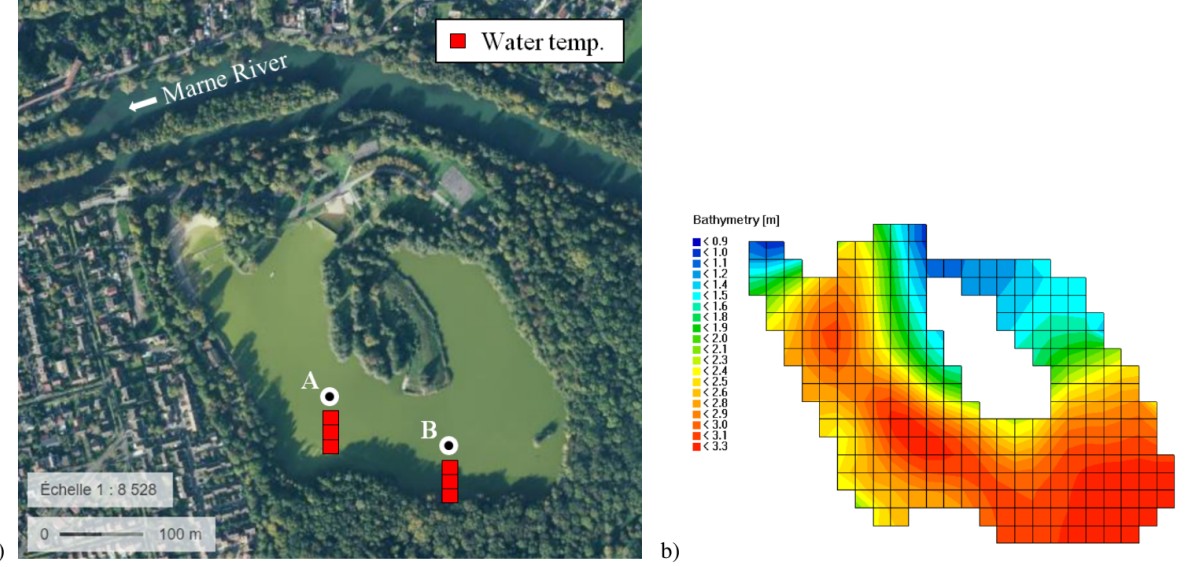

**Figure 1.** a) satellite picture of Lake Champs-sur-Marne (source: *geoportail.fr*) and sketch of the measuring system at the two locations (A and B); b) bathymetry and horizontal mesh of the study site as used in Delft3D.

and reservoirs (Piccolroaz et al., 2019; Chanudet et al., 2012; Soulignac et al., 2017). It solves the Reynolds averaged Navier-Stokes equations for an incompressible fluid under the shallow water and the Boussinesq assumptions. The time integration of the partial differential equations is done through an Alternate Direction Implicit method (Deltares, 2014; Leendertse, 1967). For the spatial discretization of the horizontal advection terms the Cyclic scheme was selected (Stelling and Leendertse, 1992).

The bathymetry and the two-dimensional mesh of the domain representing the study site are shown in Fig. 1-b. The surface of the lake is divided in 255 20 m $\times$ 20 m cells. The Z-model was implemented for the discretization of the vertical axes, with 12 fixed parallel horizontal layers of 30 cm thickness. It is generally accepted that horizontal layers help avoiding artificial mixing, improving model results in terms of thermal stratification (Hodges, 2014). Turbulent eddy viscosity and diffusivity were computed through the $k$-$\varepsilon$ turbulence closure model. Background values were set to zero [m$^2$.s$^{-1}$] for vertical viscosity and diffusivity, while they were set to 0.01 m$^2$.s$^{-1}$, after Soulignac et al. (2017) and according to the grid size, for horizontal viscosity and diffusivity. Bottom roughness was computed through Chézy's formulation with the default value for the Chézy coefficient of 65 m$^{1/2}$.s$^{-1}$.

The computation of the heat exchange at the air-water interface is done through Murakami's model (Murakami et al., 1985). It requires as input time series of relative humidity [-], air temperature [°C], net solar radiation [J.s$^{-1}$.m$^{-2}$], wind speed [m.s$^{-1}$] and direction [°N], as well as constant values for sky cloudiness [-] and Secchi depth [m]. The heat flux model computes the heat budget at the air-water interface by taking into account the net incident solar radiation ($Q_s$), the heat losses due to back radiation (long wave, $Q_b$) and evaporation (latent heat flux, $Q_e$), and the sensible convective heat flux ($Q_c$). The total upward

heat flux through the air-water interface ($Q$) is therefore:

$$Q = -Q_s + Q_b + Q_e + Q_c \tag{1}$$

Finally, evaporative mass flux is here neglected and water volume and depth are therefore considered as constant. This assumption makes it possible to analyze exclusively the impact of changes in the climatic forcing.

### 2.2.2 Meteorological input data

The meteorological forcing for this study comes from the spatialized SAFRAN (Système d'Analyse Fournissant des Renseignements Atmosphériques à la Neige) meteorological analysis system (Durand et al., 1993). SAFRAN is part of the SAFRAN-ISBA-MODCOU chain of reanalysis that covers the hydrological cycle over France, from meteorology to snow and ice formation to hydrology, respectively (Habets et al., 2008). SAFRAN integrates spatialized data from meteorological models with various sources of observations through data assimilation techniques, in order to create a consistent and spatially detailed record of meteorological data over the french territory. Its outcomes have been thoroughly validated against observed series (Quintana-Seguí et al., 2008), and tested as inputs to hydrological models (Raimonet et al., 2017). The data are spatialized on a regular square grid (8 km between each cell center) that covers the entire French Territory. The location of Lake Champs-sur-Marne falls midway on the axis connecting the centers of SAFRAN cells number 1457 (North of the lake) and 1566 (South of the lake). The average of these two cells was therefore considered representative of the conditions over the study site and used as input for the hydrodynamic model.

Data were downloaded from the SAFRAN suite in terms of: air temperature [°C], specific humidity [-], solar radiation (direct and diffused) [W.m$^{-1}$] and wind speed [m.s$^{-1}$]. All these variables are well reproduced by SAFRAN (Quintana-Seguí et al., 2008). Data were downloaded at a hourly time step, in order to accurately simulate the daily variability of the thermal profile and improve the model performance. This is crucial in shallow water bodies, where thermal stratification and mixing can alternate between day and night. Specific humidity (SH) data had to be converted into relative humidity (RH) to match the input data set needed by Delft3D. This was done through the following formula:

$$RH = 100 \cdot \frac{w}{w_s} \approx 100 \cdot \frac{SH}{w_s} \tag{2}$$

where $w$ is the mixing ratio of water with dry air [$kg.kg^{-1}$], the subscript $s$ stands for saturation conditions and SH is the specific humidity, numerically very close to the mixing ratio value. The saturation mixing ratio can be calculated as follows:

$$w_s = \frac{R_a}{R_v} \cdot \frac{e_s}{p_{atm} - e_s} \tag{3}$$

where the atmospheric pressure ($p_{atm}$) was considered to be constant and equal to the global average: $p_{atm}$ =1013 hPa. The ratio between the air and vapor ideal gas constants ($R_a$ and $R_v$, respectively) is equal to 0.622. The partial vapor pressure at saturation ($e_s$) is temperature dependent and can be estimated (in hPa) through the Magnus equation:

$$e_s = 6.1094 \cdot exp\left(\frac{17.625 \cdot T}{T + 243.04}\right) \tag{4}$$

where $T$ is air temperature [°C]. The numerical coefficients in Eq. (4) were issued from Alduchov and Eskridge (1997). Finally, in order to complete the set of meteorological input for Delft3D, daily wind direction data were downloaded from the closest available MétéoFrance station (ID: 78621001 located in Trappes, roughly 40 km West of the study site), through the INRAE CLIMATIK platform (https://intranet.inrae.fr/climatik/, in French) managed by the AgroClim laboratory of Avignon, France.

### 2.2.3 Calibration and validation

Delft3D-FLOW stands on a robust mathematical and physical structure and only few parameters have to be calibrated. Here, only those directly involved in the heat-flux model and in the wind module were calibrated: the Secchi depth [m], the mean cloud cover [-] and the wind drag coefficient [-]. The Secchi depth ($H_S$) is the parameter that defines water transparency. It is correlated with the penetration of solar radiation in water through the light extinction coefficient ($\gamma = 1.7/H_S$ (Poole and Atkins, 1929)) and therefore has a strong influence on the stratification of the water column. In order to get a first estimate for the sky cloudiness parameter, cloud cover data from the MétéoFrance station in Trappes (ID: 78621001) were averaged over the calibration period. The wind drag coefficient was calibrated in order to take into account the presence of tall trees surrounding the contour of the lake, locally reducing wind speed. The calibration was done through a trial and error procedure based on high-frequency water temperature data at the surface, middle and bottom layers (0.5, 1.5 and 2.5 m depth, respectively) during the year 2016. The model was then run for validation over the period during which both meteorological data and *in situ* observations were available, i.e. from the 15th May 2015 to the 31st December 2017.

Model results were compared to water temperature data at three depths (surface, middle and bottom of the water column) and two different locations (A and B). The root mean square error (RMSE) was calculated to evaluate model performances. For this purpose, high-frequency data were first averaged every hour to match the model output time step and cleaned from the outliers originated by periodic sensor maintenance. The latter were defined as sudden water temperature variations (> 1°C) over the 10 minutes separating two successive measurements, and consequently erased.

### 2.3 Indices for the characterization of the lake thermal regime

The thermal regime of the lake is assessed directly through the analysis of model results in terms of water temperature and through a series of indices that explore the phenology of stratification and highlight the relation between temperature and cyanobacteria production, which are described here-after. All indices are computed both on an annual and on a seasonal basis, according to the following definitions for the four seasons: (i) January, February and March (winter), (ii) April, May, June (spring), (iii) July, August, September (summer), (iv) October, November, December (autumn).

### 2.3.1 Stratification indices

In order to thoroughly characterize the phenology of stratification in Lake Champs-sur-Marne, two indices for the stability of the water column have been calculated: the Schmidt stability index and an index based on temperature difference between surface and bottom layers. The Schmidt stability index is a parameter often used in limnological studies to estimate the resistance

of a water body to mixing, and therefore its stability. It has been extensively used in scientific literature to describe the strength of stratification in lakes and, more recently, to analyze its evolution over time in relation to climate change (Vinçon-Leite et al., 2014; Niedrist et al., 2018; Kraemer et al., 2015; Livingstone, 2003) and algal blooms (Wagner and Adrian, 2009). The Schmidt stability index ($S$) represents the amount of work per unit area that would be required to mix the lake water column at one time instant. It has been here calculated following Idso's formulation (Idso, 1973), in which the vertical axis $z$ is considered positive downwards from the surface to the maximum lake depth $z_M$ [m]:

$$S = \frac{g}{A_0} \int_0^{z_M} (z_v - z)(\rho_i - \rho_v) A(z) dz \qquad [J.m^{-2}] \qquad (5)$$

where:

$$z_v = \frac{1}{V} \int_0^{z_M} z A(z) dz \qquad (6)$$

is the depth of the center of volume of the lake, $\rho_v$ [kg.m$^{-3}$] is water density at the depth of the center of volume $z_v$, $\rho_i$ is the mean uniform density, g [m.s$^{-2}$] is the acceleration of gravity, $V$ [m$^3$] and $A_0$ [m$^2$] are respectively the volume and the surface area of the lake, and $A(z)$ is the surface of the horizontal section of the lake at depth $z$. Computed for each time step, the Schmidt stability can also be averaged over each year or season.

Water resistance to mixing as estimated by the Schmidt stability index is closely correlated to temperature stratification. However, universal thresholds for the onset and breakdown of stratification are difficult to define based on this index and cannot be found in the literature, especially for shallow polymictic lakes. In order to assess the succession of stratification events in a polymictic water body, after Kerimoglu and Rinke (2013) and Magee and Wu (2017), the lake was considered to be stably stratified during a day if the minimum of $\Delta T$ is greater than 1 °C. This allows to identify all stably stratified days (SSD) and to compute their total number over a year (annual SSD), or over a season (seasonal SSD), as defined in section 2.3.

### 2.3.2 Growth rate and growing degree days

Changes in the thermal regime might impact biomass production. Here, we make use of two indices as proxies of the potential growth of phytoplankton species: the thermal growth rate (GR) and the growing degree days (GDD).

Under the assumption of nutrient and light availability, phytoplankton growth rate can be modelled, for different species, as a function of temperature as follows (Bernard and Rémond, 2012) :

$$k(T) = k_{opt} \frac{(T - T_{max})(T - T_{min})^2}{(T_{opt} - T_{min})[(T_{opt} - T_{min})(T - T_{opt}) - (T_{opt} - T_{max}(T_{opt} + T_{min} - 2T))]}, \ \forall \, T \in [T_{min}, T_{max}] \qquad (7)$$

where k$_{opt}$ is the optimal growth rate, T$_{min}$ the minimal temperature, T$_{opt}$ the optimal temperature and $T_{max}$ the maximal temperature. The model parameters were calibrated by You et al. (2018) through experimental data to describe the response

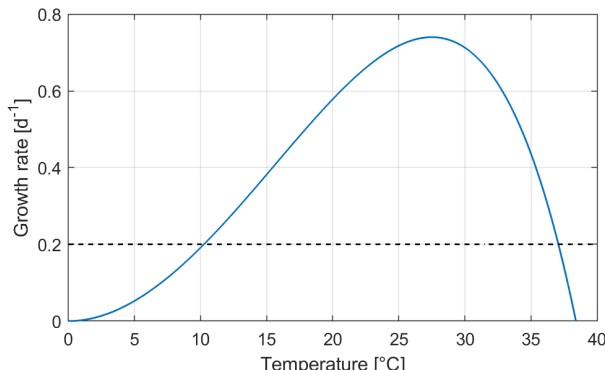

**Figure 2.** Thermal growth rate calculated after equation 7. The horizontal dashed line for GR=0.2 d$^{-1}$ meets the curve at the temperature limits for the calculation of the GDD (10°c and 37°C, respectively).

to water temperature of *Microcystis aeruginosa*, a species of cyanobacteria present in Lake Champs-sur-Marne and often dominant in freshwater bodies globally. The same values are used in this work:

$$k_{opt} = 0.74\text{d}^{-1}, \ T_{min} = 0°C, \ T_{opt} = 27.5°C, \ T_{max} = 38.4°C. \tag{8}$$

*Microcystis aeruginosa* is thought to be favored by the warmer water temperatures induced by climate change. However, the curve obtained from eq. 7 and 8 (shown in figure 2), is here more generally intended to be representative of the typical thermal response of cyanobacteria with high optimum temperature. Mean annual and seasonal (according to 2.3) growth rates are calculated through eq. 7 using simulated surface water temperature, and analysed over space and time.

The growing degree days are a weather based indicator for biological growth, widely used in the field of agronomy. Based on air temperature, it gives an estimate of the rate of development and of the span of the growing season for terrestrial plants and insects. It is a useful indicator capable to link global warming and biology (Grigorieva et al., 2010; Schlenker et al., 2007). Approaches based on GDD have been increasingly applied to phytoplankton communities and fisheries (e.g. Gillooly, 2000; Neuheimer and Taggart, 2007; Ralston et al., 2014; Dupuis and Hann, 2009), in order to correlate water temperature and phytoplankton growth while taking into account interannual variability. After Dupuis and Hann (2009), GDD were calculated as follows:

$$GDD(t) = \sum_{i=t_0}^{t} a_i \cdot (T_i - T_{base}) \cdot \Delta t, \quad \text{with } a_i = \begin{cases} 1 & \text{if } T_{base} < T_i < T_{sup} \\ 0 & \text{elsewhere} \end{cases} \tag{9}$$

where $t$ is the time (here in days) with $t_0$ the reference day to start the calculation, $\Delta t$ is the time step (equal to 1 day), $T_i$ is the daily average of the modeled surface water temperature of day $i$ and $T_{base}$ (respectively $T_{sup}$) is a physiological threshold below which (respectively above which) growth does not occur. Compared to the formulation found in Dupuis and Hann

(2009), an upper limit for growth was introduced here ($T_{sup}$) to take into account high temperature stress. Our focus here is, as for the GR, on cyanobacteria. After Thomas et al. (2016) and based on the latitude of the study site, we set the base temperature at 10°C and the upper limit for growth at 37°C. This results in considering, for the cacluation of the GDD, only temperatures that yield to a GR above 0.2 d$^{-1}$ (see figure 2).

GDD can be calculated on an annual or a seasonal basis by adjusting the values of $t_0$ and $t$. Annual GDD are calculated from the first of January until the 31st of December. Seasonal GDD are obtained according to the definitions of section 2.3.

## 2.4 Long-term analysis

In the present paper we hindcast the long-term dynamics of a small and shallow urban lake between 1960 and 2017, in order to test the influence of climate change on such ecosystems.

### 2.4.1 Long-term trends

The long-term hydrodynamic simulation starts on the first of January 1960. No data were available to set the initial conditions of the model, neither in terms of water temperature, nor in terms of current velocities. However, the model is strongly driven by the meteorological data and the influence of the initial condition vanishes after only a few days (Piccolroaz et al., 2019). Indeed, small perturbations in water temperature initial conditions ($\pm$ 2°C) were tested and resulted to vanish in 5 to 7 days. The model was therefore initialized with water at rest and with a uniform water temperature of 7°C, the average of the water temperature recorded on the lake on the first of January in 2016, 2017, 2018 and 2019. Model results are stored at a hourly time step on every element of the mesh.

Model results at site A are analysed on an annual and seasonal basis for long-term trends, in terms of water temperature (averaged over the water column) and through the indices defined in section 2.3. The presence of long-term trends is tested (with a threshold for significance $\alpha = 0.05$) through the Mann-Kendall test (Mann, 1945; Kendall, 1975), a non-parametric test for the individuation of overall monotonic trends performed here through the MATLAB software (Burkey, 2020). The Mann-Kendall test is often preferred to simple linear regression in the analysis of meteorological and hydrological time series, as it does not require any assumption on the distribution of the analysed dataset (Tímea et al., 2017; Wang et al., 2020). Once a trend is detected, its strength is evaluated through the Sen's slope estimator, that uses a linear model to evaluate the intensity of the trend (Sen, 1968).

Meteorological forcing is crucial for this work, as it drives the hydrodynamic model and represents the only source of variability in our modelling configuration. The presence of long-term trends in the meteorological dataset was also evaluated by applying the Mann-Kendall test and the Sen's slope estimator to their annual averages.

### 2.4.2 Spatial analysis

The long-term evolution and the spatial variability of the thermal regime of Lake Champs-sur-Marne was further analysed exploiting the three-dimensional model simulations. Mean annual surface water temperature, annual SSD, mean annual GR

and annual GDD were computed on the whole computational domain, with the objective of investigating the relation between climate change and time evolution of the spatial distribution of these variables. For each variable $x$, the overall mean annual value $x_m$ (averaged over the complete domain) and the deviation from the mean value $(x - x_m)$ have then been computed. In order to quantify the spatial heterogeneity of these variables, the probability distribution of the deviation from the mean value of each variable was finally calculated on the computational domain and fitted, for each year, with a non-parametric Kernel probability distribution through the Matlab *pdf* function. The resulting probability density function (PDF) was plotted over time as a heat map and the mean value as a simple line plot. This allows to visualize both the time and the spatial evolution of the variable under consideration, by looking at the mean value and at the range of values characterized by a non-zero probability.

During stably stratified periods, cyanobacteria are favored over other algal groups because of their ability to move within the water column and possibly float towards the water surface (Humphries and Lyne, 1988; Wagner and Adrian, 2009; You et al., 2018). For this reason, the spatial analysis of the GR and GDD was completed, by calculating these two indices only on stable stratified days during each year. The obtained GR were further averaged for each cell over the local number of stably stratified days. Cells that showed an annual number of SSD<10 where discarded from this analysis. Finally, the resulting modified indices were analysed over space and time as described above by using a non-parametric Kernel probability distribution as an approximation of the PDF for each simulated year.

## 3  Results

### 3.1  Model calibration and validation

The model was calibrated on the year 2016 and validated on two other periods: from May to December 2015, and during the whole year 2017. Field values for the Secchi depth in Lake Champs-sur-Marne vary between 0.5 and 3 m; using this range, the Secchi depth parameter was calibrated and finally set to 1.2 m. Sky cloudiness was calibrated and set to 80%, and a uniform wind drag coefficient was set to 0.005 [-].

Model performance during calibration and validation is shown in figure 3 relatively to site A. Parity diagrams between observed and simulated water temperature are plotted for the surface, middle and bottom layers (see panels a, b and c, respectively) and show an excellent agreement between observations and model results. A slight underestimation of surface water temperature can be noticed for the surface layer during the colder winter months, as well as a slight overestimation of the highest values of water temperature by the model, especially for the middle and surface layers (see also Fig. 3-d). However, overall model performances are satisfactory for all three layers, with RMSE values between simulated and observed water temperature of 0.85°C, 0.78°C and 0.81°C at site A during calibration, respectively for the surface (0.5 m), middle (1.5 m) and bottom (2.5 m) layers. Model results are spatially robust and satisfactory also for the validation periods, with similar RMSE values for sites A (surface:1.0°C, middle:0.96°C and bottom:0.96°C) and B (surface:1.0°C, middle:0.96°C, bottom:0.99°C).

Furthermore, the observed (blue) and simulated (orange) temperature difference between the surface and bottom layers is plotted in figure 3-e, with a dashed lined representing the 1°C threshold for the definition of the SSD. Panels f and g of figure 3 show the succession of stable stratification events as defined in section 2.3.1 calculated through observations and model sim-

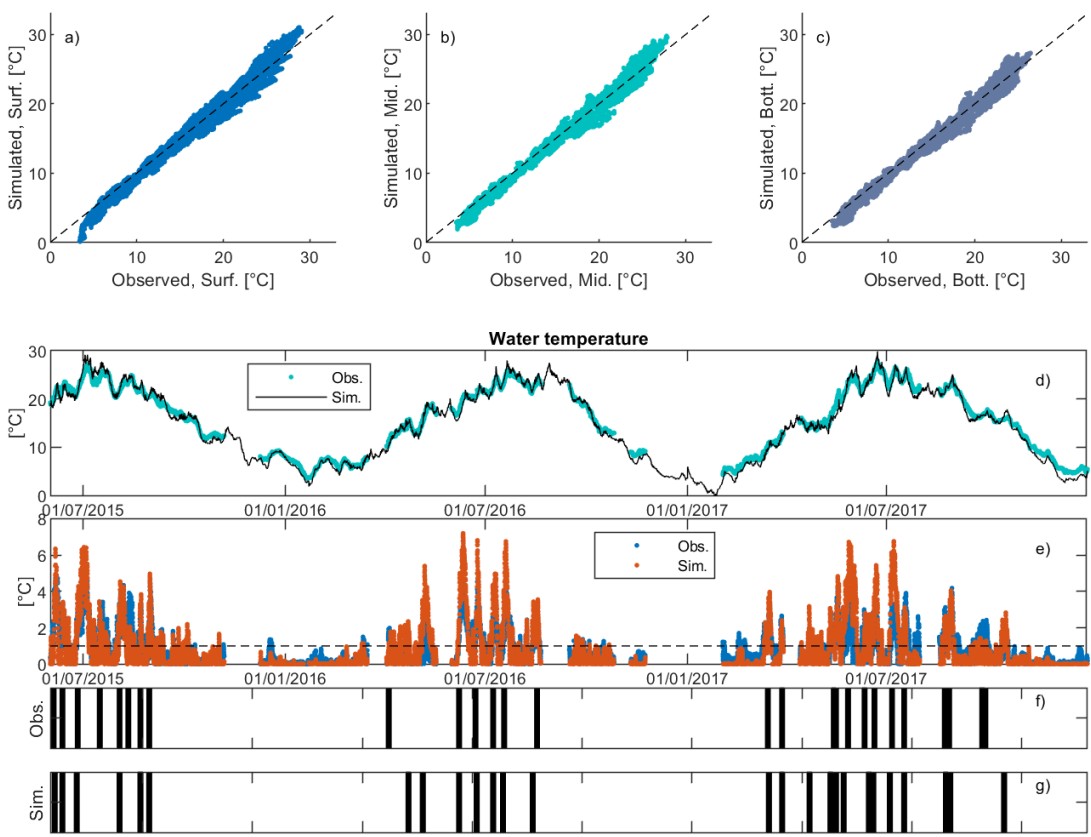

**Figure 3.** Model performance during validation at site A. Panels a, b and c: parity diagrams between simulations and observations for the surface, middle and bottom layers, respectively. Panel d: visual comparison of simulated and observed water temperature at the middle layer. Panel e: modeled (orange) vs. observed (blue) temperature difference between surface and bottom layer and relative comparison between the timing of observed and modeled stable stratification events (panels f and g, respectively).

ulations, respectively. Some discrepancy is present, notably in spring 2016, which can be explained by a slight overestimation of surface temperature, combined with the threshold effect of the definition of SSD. However, the model correctly captures the succession of stable stratification events both in terms of frequency and timing over the considered three-years period.

Overall, the model results fit very well the high-frequency water temperature data, and accurately reproduce the water temperature dynamic, including the diurnal cycle, as well as the stratification regime.

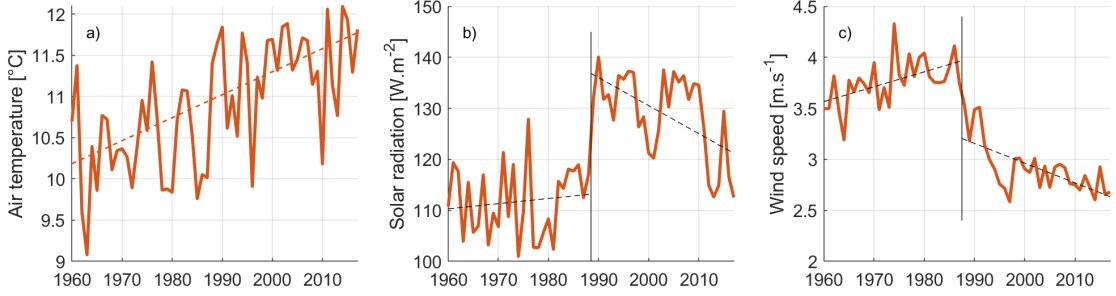

**Figure 4.** Annual averages of the three meteorological variables which exhibit significant monotonic trends, that is a) air temperature, b) solar radiation, c) wind speed. The relative overall trend intensity has been evaluated through Sen's slope estimator for air temperature (orange dashed line, panel a) whereas a piecewise trend has been calculated after change-point detection for solar radiation and wind speed (black dashed lines, panels b and c).

## 3.2 Long-term trend analysis

### 3.2.1 Meteorological input data

Annual averages of the SAFRAN dataset used as input to the Delft3D model were calculated from 1960 to 2017 and tested with the Mann-Kendall test. Strongly significant monotonic trends ($p \ll 0.05$) were found for the air temperature, solar radiation and wind speed, as shown in Fig. 4. The Sen's slope estimator was used to test the intensity of the significant monotonic trends. Air temperature displays a considerable warming trend of $0.3°C.dec^{-1}$; solar radiation also shows a significant increasing trend, with an overall intensity of $3.5$ $W.m^{-2}.dec^{-1}$. Wind speed decreases quite sharply over time, at an overall estimated rate of $0.2$ $m.s^{-1}.dec^{-1}$. While the increase in air temperature appears extremely linear (see Fig. 4-a), a sharp shift in the behavior of both solar radiation and wind speed appears around the year 1988 (Fig. 4-b and -c, respectively). A change-point detection was therefore performed on the latter two series, and showed for both variables the existence of two significant sub-trends separated by a drastic shift towards the end of the 1980s. Both variables are characterized by a mild increase until 1987 (1988 for solar radiation), followed by a considerable decrease until the end of the available series. However, despite this piecewise linear behavior, the presence of overall monotonic increasing (for solar radiation) or decreasing (for wind speed) trends is confirmed by the very low p-values obtained for these variables through the Mann-Kendall test.

Finally, no significant trend was found for relative humidity and wind direction. The two variables appear to be stationary, the former fluctuating around an annual average of roughly 80% and the latter around an annual prevailing wind direction of 200°N (South-West). Three of the five meteorological variables forcing the hydrodynamic model were therefore characterized by strongly significant monotonic trends along the past six decades, conferming changes in the climate of the region around the study site.

### 3.2.2 Model results

Long-term monotonic trends have been researched at site A on an annual and seasonal basis for: mean water temperature (vertically averaged), number of stably stratified days (SSD), mean Schmidt stability index, mean growth rate (GR) and growing degree days (GDD). Figure 5 shows all the significant monotonic trends found from this analysis. On an annual basis, the Mann-Kendall test highlighted the presence of strongly significant increasing trends ($p \ll 0.05$) for all variables.

Mean annual water temperature shows a very sharp warming tendency of $0.6°C.dec^{-1}$ (see Fig. 5-a), even greater than what was found for air temperature ($0.3°$ C). The Pearson correlation coefficient ($r$) was calculated between water temperature and the five meteorological input variables in terms of annual averages in order to explain this behavior. Modeled water temperature is strongly correlated with air temperature, solar radiation and wind speed, with correlation coefficients of 0.8 for solar radiation and air temperature and -0.9 for wind speed. Water temperature shows significant increase during all seasons, with higher slopes during spring and summer (0.8 and $0.7°C.dec^{-1}$, respectively), and a lower yet considerable intensity during autumn and winter (respectively 0.4 and $0.5°C.dec^{-1}$).

The warming trend is accompanied by reinforced stratification. An increase in water column stability is highlighted on an annual basis by both stratification indices: the annual number of SSD increased on average of around two days per decade, while the Schmidt stability index increased of $0.9$ J.m$^{-2}$.dec$^{-1}$ (Fig. 5-b and -c, respectively). Despite a warming trend being present in all seasons, both stratification related indices show significant increasing trends only during winter (1 d.dec$^{-1}$ and 0.4 J.m$^{-2}$.dec$^{-1}$) and spring (sharper trends of 1.8 d.dec$^{-1}$ and 2.6 J.m$^{-2}$.dec$^{-1}$, for the seasonal SSD and the Schmidt index, respectively). Furthermore, the number of stable stratification events (i.e. the count of the slots of consecutive SSD during a year) was calculated to characterize the frequency of stable stratification. It did not show significant trends over time, varying between a minimum value of 8 to a maximum of 16 around an overall average of 12 stable stratification events. Similarly, the duration of the longest stable stratification event (i.e. the longest slot of consecutive SSD in a year) did not show significant trends, but a high interannual variability. It varies around an average value of 11 d, between a minimum value of 5 d and a maximum of 15 d.

The analysis of the growing degree days and of the mean growth rate shows the progressive improvement of conditions for cyanobacteria. The pattern of the mean annual GR is highly correlated to that of water temperature and shows a significant trend of 0.02 d$^{-1}$ (black line in Fig. 5-d). However, the stronger intensity of the trend for the GR during spring (0.03 d$^{-1}$) indicates an amplified effect of water temperature on the potential growth of cyanobacteria during this season. Annual GDD (see figure 5-e) shows a considerable increasing rate of 157°C.d.dec$^{-1}$, with a strong shift around the year 1989. This behavior cannot be regarded as linear and is highly influenced by the piece-wise behavior of mean annual solar radiation and wind speed. However, it corroborates the idea of a greater amount of thermal energy reaching the ecosystem, at different rates but consistently throughout the four seasons.

The changes in the meteorological forcing clearly had an impact on the dynamics of the study site. The lake has sensibly warmed, its tendency to thermal stratification has increased, and the thermal conditions for cyanobacterial growth have im-

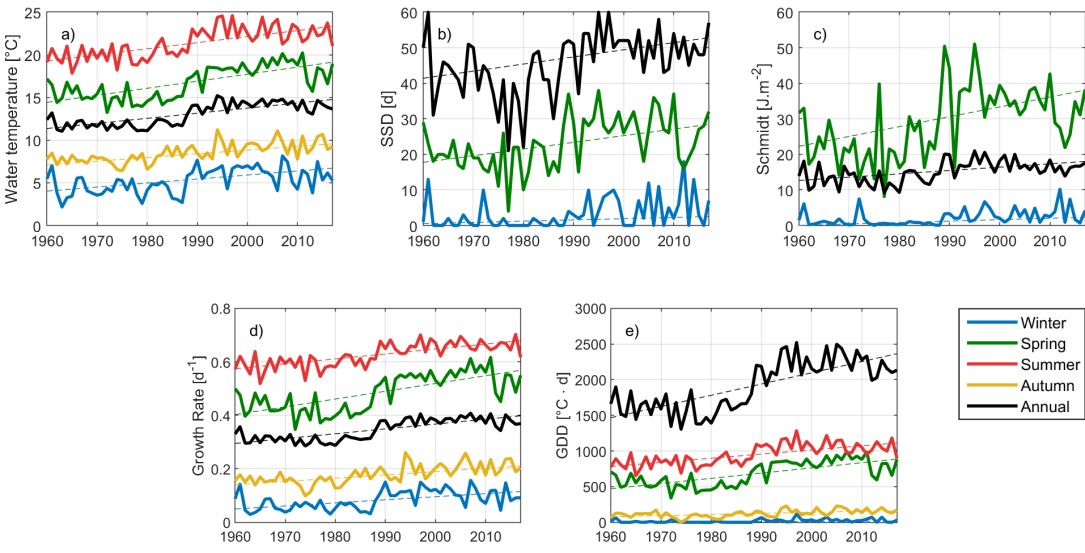

**Figure 5.** Statistically significant climate change trends at monitoring site A for the five indices, both on an annual (black) and seasonal (other colors) basis. a) Water temperature (averaged on the water column); b) Number of stably stratified days (SSD); c) Schmidt stability. d) Growth rate; e) Growing degree days (GDD). Blue lines represent the winter season, green lines represent spring, red lines are for summer trends and yellow lines for autumn; black lines represent annual values.

proved. Spring shows the sharpest trends for all indices, and might ultimately be the season suffering the strongest changes in
terms of biomass production and algal blooms.

### 3.3   Spatial analysis

Lakes are not spatially homogeneous systems. Heterogeneity can be generated by the interplay between bathymetric and morphological features, or by particular meteorological conditions, especially in terms of wind direction.

In order to quantify the rate of spatial variability in the lake, the deviations between local annual values (calculated for each
computational cell) and their overall annual mean value (calculated on the complete domain) were calculated and fitted with a probability density function (PDF). As shown in figure 6-a (top panel), mean annual surface water temperature is rather uniform over the study site. The difference between the maximum and minimum values if of roughly 0.1°C (around 1% of variability relative to the overall mean) and does not vary substantially over time. During the first half of the simulation period, and in particular between the years 1967 and 1987, the support for the PDFs (i.e. the domain on which PDFs are greater than
0) is narrower, reflecting a higher annual spatial uniformity than what can be observed after 1990. After 1990, the support of PDFs is indeed wider, with only a few exceptions where the PDFs are on the contrary quite sharp. This change in the spatial

distribution of annual surface water temperature before and after 1990 is accompanied by a sharp increase in the overall mean value (bottom panel in Fig. 6-a), which is indeed greater (around 14.5°C) after 1990 than before (around 12°C).

The annual number of SSD shows greater spatial heterogeneity (see Fig. 6-b). The difference between the maximum and minimum values of SSD varies between approximately 45 and 90 days. The spatial heterogeneity is mainly induced by bathymetry. Stable stratification only occurs in the deeper portion of the basin, while the northern part of the study site, namely the portion with depth lower than 1.8 m (see Fig. 1-b), remains constantly mixed according to our definition of SSD. The PDF is dissymmetric, with the most probable value for the annual SSD higher than the overall annual mean, by 10 to 15 days. As for the surface water temperature, the spatial heterogeneity of SSD is higher after 1990 than before. In fact, a rather high correlation is present between the spatial distribution of mean annual surface water temperature and SSD. The correlation coefficient between the two variables in each simulated year varies between 0.4 and 0.8, with an overall mean of 0.62 and p-values always lower than the threshold for significance ($p$=0.05). This suggests that surface water temperature tends to be slightly warmer in areas characterized by longer periods of stable stratification.

The thermal growth rate and the GDD were analysed over the domain during stably stratified days, which are particularly favourable to the growth of cyanobacteria.

The thermal GR shows a low spatial heterogeneity that does not vary over time, as confirmed both by the PDF in figure 7-a (top panel) and by the maps in figure 7-b. The difference between minimum and maximum values for the GR is around 0.03 d$^{-1}$ (around 5% of the overall mean value), always rather centered around the overall mean. Calculated during stratification, the overall mean thermal GR takes high values (around 0.6 d$^{-1}$), comparable to those found at site A for the summer season (see the bottom panel of fig. 7-a and fig. 5-d).

The overall mean annual GDD increases over time (bottom panel of Figure 7-c), from around 400 d.°C before 1980 to 650 d after. The PDF of the GDD displays a clear increase in spatial heterogeneity (Fig. 7-c). Its range increases substantially starting from the 1980s, roughly doubling: from 100 d.°C before 1980 to around 200 d.°C afterwards. This is due to the concurring effects of warmer water temperature and higher number of stably stratified days in the calculation of the GDD as defined in section 2.3.2. In particular, part of the heterogeneity is induced by shallow areas of the water body that only account for a low number of SSD and therefore for low values of GDD. The corresponding computational cells, not affected by stable stratification during the 1960s, are evermore likely to show stable stratification in the 2000s (see the maps in fig. 7-d). However, the maps for the years 2017 and 2005 also show a high heterogeneity in the deeper part of the water body.

## 4 Discussion

In the present paper, the thermal regime of a shallow urban lake was reconstituted over six decades (between 1960 and 2017) with a 3D thermal-hydrodynamic model. Simulation results were analysed over time (for long-term monotonic trends), and space (for spatial heterogeneity), through a series of indices that characterize the stratification and highlight the relation between temperature and cyanobacteria production.

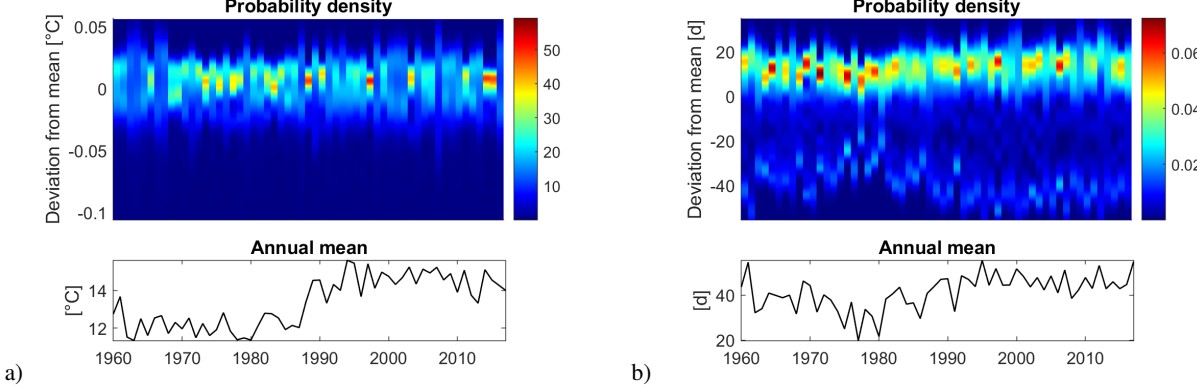

**Figure 6.** Top panels: Time evolution of the probability density function of the anomalies (i.e. the spatial deviations of a variable to its annual mean over the lake). Bottom panels: Time evolution of the annual mean calculated over the lake. a) Mean annual surface water temperature; b) annual SSD.

## 4.1 Meteorological forcing data

The model was forced with data from the SAFRAN meteorological reanalysis. Air temperature and solar radiation showed increasing monotonic trends (0.3°C.dec$^{-1}$ and 3.5 W.m$^{-2}$.dec$^{-1}$, respectively), while wind speed showed a decreasing monotonic trend of -0.2 m.s$^{-1}$.dec$^{-1}$. A shift was observed during the studied period around 1987, especially in the data series of solar radiation and wind speed, and was confirmed by a change-point detection analysis. The existence of such shift in global climate during the 1980s has been highlighted by a number of studies using different data sources (Reid et al., 2016; Mariani et al., 2012; Gallagher et al., 2013).

Climate change in the Paris region has been assessed in literature mainly in terms of air temperature (Perrier et al., 2005; Lemonsu et al., 2013). Compared to our result, a milder increasing trend of 0.1°C.dec$^{-1}$ was found based on ground measurements, between 1900 to 1987, with a steeper increment of 0.7°C.dec$^{-1}$ later on until 2005 (Perrier et al., 2005). Similarly, we also find a steeper trend of 0.55 °C.dec$^{-1}$ on the years from 1987 to 2005. Less information can be found in literature for solar radiation and wind speed. A decrease in wind speed on land was found over Europe since 1980 (around -0.1 m.s-1.dec$^{-1}$) as part of a large-scale analysis of observations in the northern hemisphere (Vautard et al., 2010). At a global scale, an overall decreasing trend in wind speed was found over land in the period 1985-2015 through meteorological reanalysis, principally over Europe, India and western Africa (Torralba et al., 2017). In South-East China, in the Lake Chaohu region, a strong decline in wind speed (China Meteorological station) was also found in the period 1980-2016 (Zhang et al., 2020). An overall increase in surface solar radiation was recently found for Europe between 1983 and 2015, specifically of 3 W.m$^{-2}$.dec$^{-1}$ for western Europe (Pfeifroth et al., 2018).

Meteorological reanalyses usually cover multi-decadal periods and have the great benefit of being spatialized over vast portions of the globe. Even though their use in limnological studies is quite recent, they have already been used to simulate

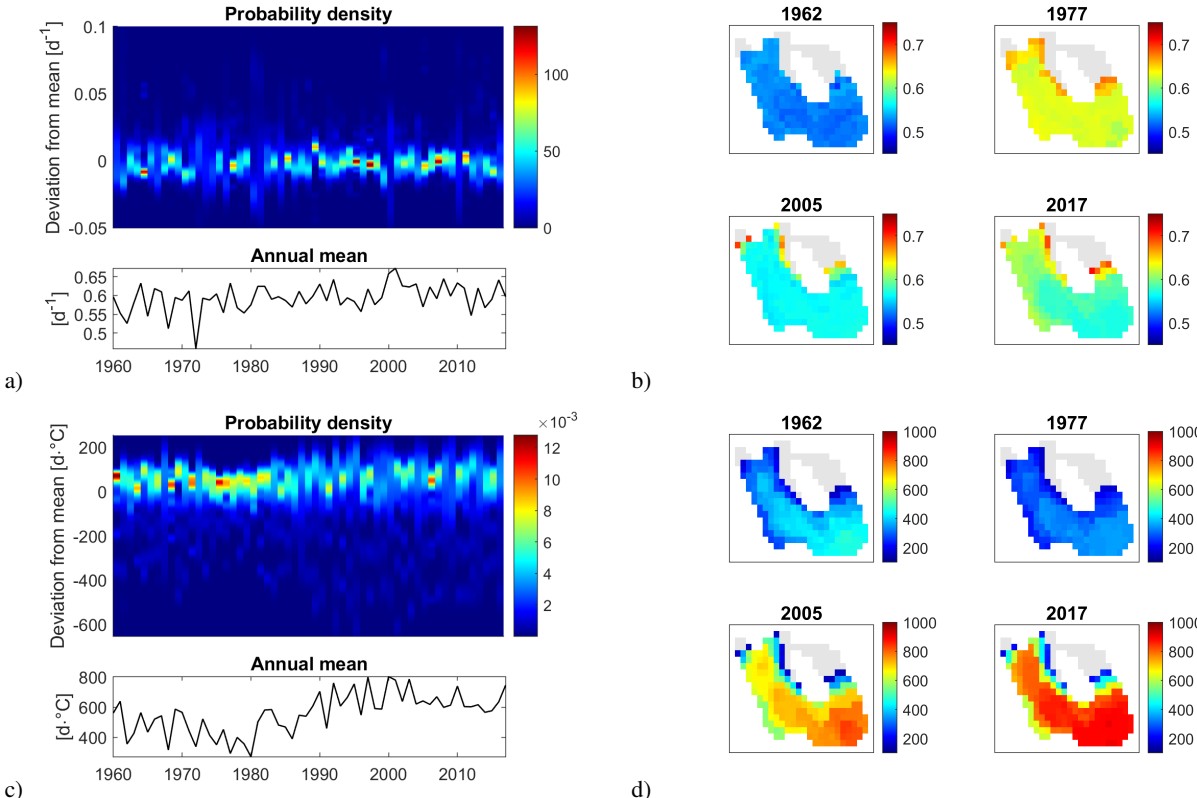

**Figure 7.** Spatial analysis of stratification. a) Probability density function (PDF) for mean GR during stratification over the computational domain and over time; b) Four examples of spatial distribution for mean GR during stratification over the lake; c) PDF for GDD during stratification over the computational domain and over the years; d) Four examples of spatial distribution for annual GDD during stratification over the lake. Grey cells in panels b and d do not stratify longer than 10 days over a year.

water temperature (Layden et al., 2016; Piccolroaz et al., 2020), stratification dynamics (Frassl et al., 2018) and phytoplankton
distribution (Soulignac et al., 2018). As shown in this work, their use as external forcing to thermal-hydrodynamic models can
yield, provided that observations are available for calibration and validation, to accurate simulations of the behavior of water
bodies even in the absence of local meteorological observations. This could open to a great range of applications in limnology
and paleolimnology (Jenny et al., 2016; Maier et al., 2019). The proposed methodology allows to thoroughly reconstruct
the behavior of any water body both in time and space, independently of its proximity to meteorological stations. This is
particularly interesting for small or remote water bodies that often lack long-term measurements.

## 4.2   Water temperature and stratification

Based on the 3D model results found for Lake Champs-sur-Marne, long-term trends were analysed in detail at site A. Significant
increasing trends were detected for water temperature both on an annual and seasonal basis. The highest seasonal warming was

found during spring and summer (0.8 and 0.7°C.dec$^{-1}$ respectively). These trends are particularly intense and could have strong impacts on the ecosystem under examination. In particular, the intensity of these trends is greater than that suggested for summer water temperature in a large-scale analysis (0.53°C.dec$^{-1}$) for lakes with similar changes in the meteorological forcing (O'Reilly et al., 2015). Furthermore, mean annual depth-averaged water temperature also increased at a considerable rate of 0.6°C.dec$^{-1}$, greater than the rate found for air temperature, a behavior also highlighted for other water bodies (Austin and Colman, 2007; Schneider et al., 2009). The piecewise linear behavior of mean annual water temperature, is induced by that of solar radiation and wind speed. In fact, similarly to what was found by Magee and Wu (2017), mean annual water temperature was highly correlated (i.e. |r| > 0.8) with air temperature, solar radiation and wind speed. This suggests that meteorological variables have additive effects that concur to enhance the response of the dependent variables. These effects might be particularly intense for wind over small and shallow lakes, due to their low volume to surface ratio.

Both stratification related indices (SSD and Schmidt stability) showed a significant mean annual increasing trend (3 d.dec$^{-1}$ and 1 J.m$^{-2}$.dec$^{-1}$, respectively). Similar values were recently found for shallow water bodies in other long-term studies (Magee and Wu, 2017; Moras et al., 2019). However, despite a strong augmentation in water temperature, stratification did not show a significant increase during summer. In shallow polymictic lakes the water column is mixed frequently also during the warmer seasons. Summer surface and bottom water temperature increased at a very similar rate (0.7°C.dec$^{-1}$) in the study site, resulting in small changes in Schmidt stability and number of SSD. This result marks a strong difference with the behavior of deeper monomictic or dimictic lakes, where the summer Schmidt stability often shows an increasing trend (e.g. Niedrist et al., 2018; Flaim et al., 2016), but it is not uncommon for shallow water bodies, where Schmidt stability can even show decreasing summer trends (Fu et al., 2020).

Stratification induces a separation between the sediment and the surface layers, influencing the distribution of nutrients and biomass over the water column. During stratification, due to the desoxygenation of the lake bottom layers, nutrients (phosphate in particular) are released from the sediment. In polymictic water bodies, when mixing occurs, a replenishment of the whole water column with the nutrients released during previous stratification has been observed (Song et al., 2013; Wilhelm and Adrian, 2008). In Lake Champs-sur-Marne, neither the frequency nor the duration of the stable stratification events show a significant trend during the past decades. However, with a mean value of 12 annual separated stable stratification events, lasting up to two consecutive weeks, the replenishment of the water column with nutrients is ensured. The multiple pulses associated with the alternation between mixing and stratification events are an important internal source of nutrients, especially in a lake such as the study site, whose water inflow is limited to underground waters.

The thermal regime was further characterized over the computational domain by analyzing the spatial distribution of surface water temperature. While annual averages of surface water temperature are rather uniform over the domain, with around 0.1°C of difference between maximal and minimal values, the bathymetric variations induced greater variability in the distribution of SSD. The stratification regime drastically changes between the deeper portion of the water body and the shallower northern part. According to the definition of the SSD, stable stratification never occurs in cells with water depth lower than 1.8 m. In shallow water bodies, even small bathymetric variations can cause drastic differences in the thermal regime. Different regimes of mixing and stratification between shallower and deeper areas can result in considerable differences in the spatial distribution

of nutrients, with effects on bloom initiation and phytoplankton growth, as well as on the resulting oxygen concentration. However, spatial heterogeneity of the mixing and stratification regime inside a water body is rarely addressed in scientific literature, especially with regard to small and shallow lakes (e.g. Bachmann et al., 2000).

## 4.3 Indices for primary production

The thermal regime is a key factor in the regulation of the biogeochemical cycle and in the development of algal blooms. The worldwide intensification of harmful algal blooms over the past decades (Paerl and Huisman, 2008; Paerl and Paul, 2012; Wagner and Erickson, 2017) is often associated with climate change and nutrient enrichment (Zou et al., 2020; Huisman et al., 2018).

Due to their potential toxicity, cyanobacteria are of particular concern in freshwater management. Warmer water temperature can favor their growth because of their high optimal temperatures. However, they can proliferate under a wide range of temperatures (Lürling et al., 2013; Carey et al., 2012). The expression of the growth rate proposed by Bernard and Rémond (2012) (see eq. 7) accounts for this dependence from water temperature. Based on this expression, the mean annual thermal growth rate of cyanobacteria showed a significant increasing monotonic trend of 0.02 $d^{-1}.dec^{-1}$. Compared to the initial annual value of roughly 0.3 $d^{-1}$ at the beginning of the 1960s, this results in a considerable total rate of change of +40% at the end of the studied period. Significant trends were also found during the four seasons, the highest being during spring (0.03 $d^{-1}$, or +45% of the initial value). The growing degree days (GDD) of cyanobacteria were analysed here for a range of temperatures comprised between 10°C and 37°C, corresponding to thermal growth rates higher than 0.2 $d^{-1}$. However, given the temperate climate of the region under examination, the upper limit for growth did not have any effect on the results, whilst it could be an important parameter for species with lower optimum temperatures such as diatoms.

Whereas the growth rate gives an estimation of the mean value of cyanobacteria growth, that can be computed on a seasonal and an annual basis, the GDD is a cumulative index that gives a measure of the amount of time and degrees available during a year for photosynthetic growth. Originating from the field of agronomy and forestry, it represents a "thermal time" and is considered as a better descriptor of vegetal phenology than the simple Julian days (McMaster and Wilhelm, 1997). Under an appropriate temperature range, it can be considered as representative for organism developmental time (Dupuis and Hann, 2009). The highest trend for GDD was found on an annual basis (157 $d.°C.dec^{-1}$), denoting that the temperatures favourable to cyanobacteria growth are more and more frequently reached. Seasonal trends varied greatly in intensity. The highest was found for spring (73 $d.°C.dec^{-1}$) and represents, relative to the values in the early 1960s, a substantial increase of 90% during the six dacades under consideration. The trends found for winter and autumn are mild but denote an increased tendency to overpass the base temperature during these two seasons, and therefore a dilatation of the season favourable to cyanobacteria growth.

Harmful algal blooms and phytoplankton dynamics depend on factors such as the settling or buoyancy rate of phytoplankton, the availability of nutrients over the water column, which can be enhanced by the release from the sediment, and the resuspension of particulate organic matter. In polymictic water bodies, the processes of sedimentation and resuspension are strongly influenced by the alternation between mixing and stratification (Song et al., 2013). Because of their ability to migrate within the water column, stratified environments are favorable to cyanobacteria (e.g. Carey et al., 2012; Aparicio Medrano

et al., 2016). The increase of water temperature and of stable stratification could concur resulting in frequent cyanobacteria blooms. However, stratified conditions do not occur uniformly. The calculation of the thermal GR and of the GDD quantifies the potential effect of water temperature on cyanobacteria growth, under the hypothesis of nutrient and light availability. Their calculation during stratification allows to address the combined effect of water temperature during a particularly favourable environmental conditions.

During stratification cyanobacteria GR was characterized by high values (around $0.6 \, \mathrm{d}^{-1}$), with a variability quite uniform over time of $\pm 5\%$ over the study site. These values are comparable, or even higher (until the 1990s) than those obtained during the summer season. The GDD give a deeper insight on the interplay between temperature and stratification. The strong augmentation in the overall mean value of GDD during stratification confirms a concurring positive effect of the increase of water temperature and of the duration of stable stratification on the growth of cyanobacteria. Moreover, the greater spatial variability of GDD values during the second half of the simulation indicates that some parts of the lake will be more affected than others by the variation of water temperature and stratification. In particular we observe the development over time of certain areas in the study site, especially the deeper part, with very high values of GDD under stratified conditions, and that are therefore particularly favourable to cyanobacteria dominance and bloom initiation.

The combination of increasing trends for water temperature, stable stratification and the widening of the growing season can favour the occurrence of harmful cyanobacterial blooms (Winder and Sommer, 2012; Jones and Brett, 2014; Noble and Hassall, 2015). If these trends are confirmed, during the decades to come cyanobacteria could become the dominant species in the study site, seriously affecting the lake ecological network and its biodiversity (Rasconi et al., 2017; Toporowska and Pawlik-Skowronska, 2014).

## 4.4 Model-based approach

Through our modelling approach it was possible to reconstruct the thermal dynamics of a small and shallow lake and to thoroughly analyze its evolution over time and space. The use of an extensive data set of high-frequency observations allowed to test the model not only against the general seasonal water temperature pattern, but also against daily and sub-daily dynamics of stratification and mixing, at two locations. Other works have focused on the hindcast of lakes thermal regime, successfully reconstructing their dynamics in order to analyze their evolution over time (e.g. Magee and Wu, 2017; Moras et al., 2019; Zhang et al., 2020; Stetler et al., 2020). Most of these studies, however, make use of a 1D approach. By means of a 3D model it is possible to aggregate information on both time and space (horizontal and vertical) through the use of appropriate indices. Our work demonstrates that even on a small water body spatial variations can be important, and that their influence on the thermal and biological regime must be considered. It provides additional evidence that supports the hypothesis of a positive effect of climate change over cyanobacteria blooms.

Hydro- and thermal dynamics are at the core of the biogeochemical cycle, influencing transport, sediment resuspension, organic matter mineralization in addition to primary production. In this work, we focus on water temperature, quantifying its impact for stratification and biological production. The proposed methodology allows to focus solely on the role of the meteorological forcing, addressing their direct impact on the thermal regime and on primary production. However, other factors

could have even a stronger impact: nutrient and light limitation or grazing could offset temperature-derived advantages (Elliott et al., 2006). These factors are not taken into account in this work, since it is focused on the impact of climate change from a thermal standpoint, all other factors being equal. This work opens to a wide range of additional analysis and further research. In particular, the coupling with a biogeochemical model could give further insight on the impact of climate change on the ecological state of a water body. Such a study, however, would introduce additional sources of uncertainties, especially regarding the evolution of nutrient sources over time and could only be profitably if performed after a thorough analysis of the hydrodynamic and thermal regime.

## 5 Conclusions

In this work, the long-term thermal regime of a shallow urban lake is reconstructed through model simulations from 1960 to 2017. A series of indices are proposed with the objective of thoroughly describing the thermal regime of shallow water bodies, in relation with stratification dynamics and cyanobacterial production. The meteorological data set is derived from the SAFRAN reanalysis and shows a significant increase in air temperature and solar radiation and a significant decrease in wind speed, with a regime shift in the late 1980s. Simulation results show that small urban lakes react rapidly and strongly to external meteorological conditions, with only limited resilience to climatic shifts. The additive effect of increasing solar radiation and air temperature and decreasing wind speed acts on different terms of the heat budget at the lake surface, enhancing the changes found in the lake. The mean water warming of $0.6°C.dec^{-1}$ represents an increase of 32% in water temperature values between 1960 and 2017 and is much stronger than the air warming ($0.3°C.dec^{-1}$, i.e. an increase of 18% during the same period). The impact on stratification and cyanobacteria production is even more alarming, with an increase of over 30% of the stability indices and over 60% of the growing degree days during the six past decades. Spring shows the sharpest trends in terms of water temperature, water column stability (Schmidt and SSD) and growing degree days, and might ultimately be the season suffering the strongest changes in terms of biomass production and algal blooms. The spatial heterogeneity found for thermal stratification and growing degree days might also concur to locally create conditions particularly favourable for cyanobacteria blooms. These tendencies could favour early phytoplankton blooms (during late winter or spring) and contribute to the proliferation of cyanobacteria, and ultimately to the degradation of the whole aquatic ecosystem. Our results highlight the importance of a three-dimensional approach to thoroughly infer the dynamics of a water body. Horizontal patterns can be particularly strong for shallow lakes due to the relative importance of bathymetric variations.

Small and shallow lakes are extremely widespread ecosystems. Our results suggest that such systems experience considerable thermal stress caused by climate change and that, in nutrient-enriched systems, cyanobacteria dominance could become a widespread issue in the future decades.

*Code and data availability.* The model set-up for long-term simulations, as well as the corresponding results at site A are available on Mendeley (https://data.mendeley.com/datasets/92kzf5t5xn/draft?a=11918779-ce63-4e72-aa69-9207e8445fdc). Model results were obtained

using the Delft3D software package (Delft3D-flow, version 4.01.01.rc.03). Matlab codes used to obtain the datasets for this paper are available upon request from the corresponding author.

*Author contributions.* FP: conceptualization, investigation, writing - original draft. CC: conceptualization, supervision, writing - review & editing, funding acquisition, project administration. BJL: conceptualization, writing - review & editing. PLM: data curation, writing - review & editing. PD: data curation. BVL: conceptualization, supervision, writing - review & editing, funding acquisition, project administration.

*Competing interests.* The authors declare that they have no conflict of interest.

*Acknowledgements.* The data set used for model calibration and validation was collected under the OSSCyano (ANR-13-ECOT-0001) and ANSWER (ANR-16-CE32-0009-02) projects. The environmental observatories OSU EFLUVE and OLA contributed to the financial support for equipment maintenance. The authors also acknowledge the Base de loisirs du lac de Champs/Marne (CD93) for their logistical support in the field campaigns. The first author's PhD grant is funded by Ecole des Ponts ParisTech and ANSWER project. The authors also thank Nicolas Clercin for the revision of the English text.

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
