# Peer review of "The thermal response of small and shallow lakes to climate change: new insights from 3D hindcast modelling."

_Earth System Dynamics, 2020_

## Referee Comment (RC1) · Anonymous Referee #1 · 22 Aug 2020

**General comments:**

It is not a difficult task to go through this manuscript. On the one hand it is well organized and structured, which makes me quickly grasp all the important points the authors want to illustrate. However, on the other hand I have to say that all the results in this research (e.g. air temperature in the region strongly increased in the last several decades which elevated water temperature and Schimdt Stability) is not so new to me, and it gives me a feeling that if its research place changed to somewhere else then the results will be quite similar to those in some previous studies. The authors said the innovation point is that most of previous research neglected the climate change effects on small lakes, which is not true for me. In recent years there are already some research on this topic (see my comments below). For my point of view the innovation points in this paper is the application of 3D model to check influence of climate warming on lakes based on long-term simulations, and this should be reflected in the Results and Discussion (see my comments below).

To avoid the misunderstanding, I do not say this research is bad, instead I recommend the authors inputting some new findings, as suggested below, to make the manuscript more attractive.

**Detailed comments:**

**Line 29:** This sentence is not so reasonable, since recently there are quite a few studies to check the response of small lakes to climate change, like:

1. A small temperate lake in the 21st century: Dynamics of water temperature, ice phenology, dissolved oxygen and chlorophyll a

2. Future projections of temperature and mixing regime of European temperate lakes.

It is better to further modify this sentence, to make it a bit soft.

**Line 57:** Better to slightly modify it like,"Although the proposed ....., it is generic and...."

**Line 101:** I am not so sure whether it is reasonable to use "constant values for sky cloudiness". Since in this way incoming longwave radiation only depends on air temperature which may make the results not so accurate. What do you think of it??

**Line 108:** for the part 2.2.2 Meteorological input data, If you used the historical reanalysis climate data to drive the hydrodynamic model, I supposed it is better to firstly show the accuracy of this reanalysis in capturing the real climate condition. Have you compared the reanalysis results with the real

measurements??

**For Figure 2:** the legend doesn't match with the plot, please modify it.

**Line 299:** For Schimidt stability it is not quite meaningful to show the result in winter, instead it is better to analyze it in summer which should be input into Figure 4c.

**For part 3.3 Spatial analysis of stratification:** I mean this part is not new to me. It is very normal that the shallow part of a lake experiences less stratification compared to the deep part. I recommend that the authors also check the horizontal distribution of other thermal indices, like mixed layer depth, thermocline depth, which may provide some interesting findings.

**For 4 Discussion:** Just as I suggested before, results in this paper is not so new compared to those from previous studies, as well as the Discussion. The innovative points, based on my opinion, is the application of 3D model to check the influence of climate change on lakes. Consequently there should be more discussion, at the last paragraph of this part, on how the heterogeneity of thermal structure in the horizontal direction can affect the aquatic ecosystems,

---

## Referee Comment (RC2) · Anonymous Referee #2 · 23 Sep 2020

The work belongs to the growing number of case studies on lake response to the recent climate change. Here, the authors investigated the long-term trends in a small shallow artificial lake by applying a 3-dimensional hydrodynamic model driven by a 1960-2017 regional meteorological reanalysis dataset. The combination of the study object (shallow polymictic lake) and the approach (a full 3-d model) is particularly interesting for revealing the fine mechanisms and effects of the regional climate change. The results are presented in a clear and straightforward way, but the abovementioned potential of the study is barely unfold. Except one sentence in Conclusions, the motivation for application of a 3-d lake model is not discussed, neither its advantages and disadvantages are discussed compared with simpler 1d models. It remains unclear,

why should one use such a complicated model, subject to a lot of uncertainties, just to arrive at an obvious conclusion: the lake parts shallower than the mean depth of the mixed layer do not stratify. It sounds like cracking nuts with a sledgehummer. The 3d model performance is only briefly addressed. The validation was performed only on surface (mean) temperature, which is not sufficient to trust the later model results on the stratification trends. The question about the model ability to adequately reproduce vertical thermal stratification in the lake remains open.

A large part of discussion is dedicated to the effect of climate change on the transient stratification development in shallow polymictic lakes. However, the stratification indices used in the analysis—Schmidt stability and the total stratification duration—are rather relevant to oligomictic (di- and monomictic) lakes. Neither duration of the longest stratification period, nor the frequency of stratification events are analyzed. The indices used for the warming effect on the net biological production are also questionable: temerature as a measure of the growing season is weakly justified in lake ecosystems, especially for the climate under consideration. The "number of growing days" (NGD) in the authors' formulation is always clearly above 300, so the whole 365-days long year cycle can be a priori assumed as production-favorable in terms of temperature. Trends in GDD and NGD do not seem to be representative for any biotic processes. In particular, because high temperatures can work as a stress factor inhibiting both primary production and growth rates of higher organisms. In general, Discussion presents a lot of common knowledge but is weakly connected to the results from the study.

In summary: the study uses a promising approach and a solid dataset, but, in its current form, presents little advance on the subject under investigation. A stronger focus on the abilities of 3-d modeling for climatic lake studies and intermittent stratification dynamics of shallow polymictic lakes would strengthen this otherwise well-designed and clearly structured study.

Here are some specific remarks:

[Figure]

- The model uses constant cloudiness as input, which is quite strange, especialy, taking into accout significant long-term trends in solar radiation. Why the real variability of cloud cover (or long-wave atmospheric radiation) was not used? Can you estimate the resulting errors in the model output?

- Also, a constant water transparency is used in the long-term model runs, despite the data indicate a strong transparency variability on seasonal scales. How this assumption affected the model predictions on stratification patterns? Will the time-variable Secchi depth change the modeling results?

- I do not believe that the trends in solar radiation and wind are monotonic (Fig. 3bc). A change point detection analysis should be performed here (e.g. B.K. and Tsay, R.S., 2002. Bayesian methods for change-point detection in long-range dependent processes. Journal of Time Series Analysis, 23(6), pp.687-705., or any other similar approach) with subsequent piecewise trend estimation.

---

## Author Comment (AC1) · 21 Oct 2020

**Authors' response to referee 1**

**General comments:**

It is not a difficult task to go through this manuscript. On the one hand it is well organized and structured, which makes me quickly grasp all the important points the authors want to illustrate. However, on the other hand I have to say that all the results in this research (e.g. air temperature in the region strongly increased in the last several decades which elevated water temperature and Schimdt Stability) is not so new to me, and it gives me a feeling that if its research place changed to somewhere else then the results will be quite similar to those in some previous studies. The authors said the innovation point is that most of previous research neglected the climate change effects on small lakes, which is not true for me. In recent years there are already some research on this topic (see my comments below). For my point of view the innovation points in this paper is the application of 3D model to check influence of climate warming on lakes based on long-term simulations, and this should be reflected in the Results and Discussion (see my comments below).

To avoid the misunderstanding, I do not say this research is bad, instead I recommend the authors inputting some new findings, as suggested below, to make the manuscript more attractive.

> **Authors' response:** First of all, my co-authors and I would like to thank you for the interest in our work and for this review.
>
> We agree on the general consideration that, if the study site under consideration changed, the results would be close to those found in previous studies. On the first hand though, this remark adds some robustness to our modelling set up as well as to our findings. On the second hand, the main idea behind this work was not only to evaluate the 3D effects of climate change on a single small polymictic lake, but also to propose an approach able to generalize to some extent the results to the numerous similar ecosystems that fit into the similar temperate climatic conditions and morphologic characteristics. In our revised manuscript, we focus more closely on this aspect.
>
> We agree on the general remark that a stronger focus on 3D results is needed in order to highlight the novelty of the approach. This is addressed in the revised draft both in the results and in the discussion section.
>
> Concerning the novelty of our results, we do not see the innovation in this paper lying only in the application of a 3D model. Even though this is surely one of the main innovative points of the work, also the use of the SAFRAN reanalysis to hindcast the lake hydrodynamics and the use of high-frequency data for including daily cycles in the calibration and validation of the model constitute innovative elements, especially for small urban lakes. Finally, also the use of ecology-derived indices (such as the growing degree days) as proxies for the impact of climate on potential biomass growth in aquatic ecosystems adds an element of innovation.
>
> Finally, it is true that our writing did not highlight enough these additional elements of novelty, that should now be more clearly brought to the attention of the reader in the revised version.

**Detailed comments:**

**Line 29:** This sentence is not so reasonable, since recently there are quite a few studies to check the response of small lakes to climate change, like:

1. A small temperate lake in the 21st century: Dynamics of water temperature, ice phenology, dissolved oxygen and chlorophyll a

2. Future projections of temperature and mixing regime of European temperate lakes.

It is better to further modify this sentence, to make it a bit soft.

> **Authors' response:** Thank you for these references, now cited in the revised version. However, the intention behind the sentence under examination was to highlight how small polymictic lakes have received

overall less attention than, for instance, larger mono- or dimictic lakes. The paragraph was modified (see italics) into:

"On the other hand, small and *shallow lakes (i.e. with surface < 1 km$^2$, and light potentially penetrating to the bottom (Meerhoff and Jeppesen, 2009)) have received less attention in climate change impact studies than deeper monomictic or dimictic water bodies*. Nevertheless, they play an important role for biodiversity and are prone to ecological deterioration and harmful algal blooms (Biggs et al., 2016; Wilkinson et al., 2020). Furthermore, with the advance of urbanization, the presence of aquatic environments has become a key feature for the improvement of life quality in the urban landscape (Frumkin et al., 2017; van den Bosch and Sang, 2017). Often small and shallow, urban lakes grant valuable ecosystem services and contribute to the preservation of biodiversity (Frumkin et al., 2017; Hill et al., 2017; Hassall, 2014; Higgins et al., 2019). *For these reasons, in recent years small polymictic lakes gained greater attention in scientific studies.* However, to our best knowledge, only a few studies have quantified the effect of climate change on these ecosystems (*Tan et al., 2018; Shatwell et al., 2019,* Moras et al., 2019)."

**Line 57:** Better to slightly modify it like, "Although the proposed ....., it is generic and...."

> **Authors' response:** The sentence has been modified into: "*Although the proposed methodology was here applied to a study site located in the Paris region, it is generic and could be applied to other similar sites*."

**Line 101:** I am not so sure whether it is reasonable to use "constant values for sky cloudiness". Since in this way incoming longwave radiation only depends on air temperature which may make the results not so accurate. What do you think of it??

> **Authors' response:** We agree that it would be preferable to use cloudiness measurements. Cloud cover measurements were available from the closest meteorological station, 24 km away, but not for the 8x8 km SAFRAN reanalysis cell, contrary to other meteorological variables.
> The model was tested during calibration and validation using two different modules for heat exchange, the Ocean model with time series of cloud cover, and the Murakami model which is implemented with a constant cloud cover in Delft3D. Overall, water temperature was slightly better modeled by the Murakami model with a specifically calibrated value for cloud cover. For instance, for the year 2016, RMSE at site A was around 1.1°C for all three depths with the Ocean heat exchange model, while it was around 0.8°C for the Murakami heat exchange model.
> This can be explained on the first hand by the high uncertainty that affects ground-based cloud cover observations (Silva and Souza-Echer, 2015, Zelinka et al., 2017) and by some gaps found in the series, and on the second hand by the different source of these data from the other meteorological input. Furthermore, a preliminary analysis that we carried out on 19 years (from 2000 to 2018) of cloud cover data from the closest meteorological station, showed an overall very low seasonality in the series, (with 7 oktas (87%) being the most probable value for all seasons and 65% being the overall mean value) and no interannual trend. The absence of strong climatic trends for the area of interest in terms of cloud cover is confirmed by Pfeifroth et al. (2018).
> We therefore decided to use the constant calibrated value of sky cloudiness in order to achieve the best results in terms of simulated water temperature against our set of hourly averaged water temperature observations. In any case, this led to very good model performances during both calibration and validation. Finally, we think that the calibration of the parameter to a value of 80% does not introduce considerable biases in our work, even in the long term study, also given the non-negligible uncertainty that affects cloud cover observations.

**Line 108:** for the part 2.2.2 Meteorological input data, If you used the historical reanalysis climate data to drive the hydrodynamic model, I supposed it is better to firstly show the accuracy of this reanalysis in capturing the real climate condition. Have you compared the reanalysis results with the real measurements??

> **Authors' response:** A thorough validation of SAFRAN can be found in Quintana-Seguì et al. (2008), where SAFRAN is compared directly with meteorological observations. Air temperature, wind speed, relative humidity and incoming solar radiation are well reproduced by SAFRAN with small bias and high correlation with observation series. For our study site, we obtained very similar correlation and RMSE values during a preliminary comparison between SAFRAN data and time series from the closest meteorological station. In the revised draft, we explicitly mention the process of validation of the SAFRAN data around lines 116 and 123 (see italics):
>
> "SAFRAN integrates spatialized data from meteorological models with various sources of observations through data assimilation techniques, in order to create a consistent and spatially detailed record of meteorological data over the French territory. *The accuracy of the SAFRAN reanalysis has been thoroughly validated by comparison with observed data series (Quintana et al., 2008), and the reanalysis data were tested as inputs to hydrological models (Raimonet et al., 2017) with success*. Reanalysis data were downloaded from the SAFRAN suite in terms of: air temperature [°C], specific humidity [-], solar radiation (direct and diffused) [$W.m^{-1}$] and wind speed [$m.s^{-1}$]. *All these variables are very well reproduced by SAFRAN (Quintana et al., 2008).*"

**For Figure 2:** the legend doesn't match with the plot, please modify it.

> **Authors' response:** The figure under examination will change in the revised draft to allow for a deeper discussion of the calibration and validation of the model (see Figure 1 in this document). Parity diagrams between observations and simulations were added for three layers (panels a, b and c), and the timing and frequency of modeled and observed stable stratification events were also addressed. The caption and legend have changed accordingly:

[Figure]

**Figure 1:** Model performance during validation at site A. Panels a, b and c: parity diagrams between simulations and observations for the surface, middle and bottom layers, respectively. Panels d and e: visual comparison of simulated and observed water temperature at the middle layer (d) and corresponding residuals (e). Panel f: modeled (orange) vs. observed (blue) temperature difference between surface and bottom layer and comparison between the timing of observed and modeled stable stratification events.

**Line 299:** For Schimidt stability it is not quite meaningful to show the result in winter, instead it is better to analyze it in summer which should be input into Figure 4c.

**Authors' response:** Schmidt stability was analyzed during Summer but did not show any significant trend, and for this reason it was not shown in figure 4, as stated in the caption. However, it was probably unclearly explained, and it is true that we did not further comment this point in the discussion. The absence of a trend in the summer average of Schmidt stability is a relevant result, as it marks a strong difference from dimictic or monomictic lakes. We therefore extended our discussion on this matter:

"*Despite a strong augmentation in water temperature, stratification did not show any significant increase during Summer. This is due to the shallowness and polymicticity of the study site, that allow the bottom layer to be heated and the water column to be mixed frequently even during Summer. Summer surface and bottom water temperatures increased at a very similar rate over time (0.7°C.dec$^{-1}$), preventing significant changes in the Schmidt index and in the number of SSD. This result marks a strong difference with the reported behavior of deeper monomictic or dimictic lakes, where the stable seasonal stratification can induce an intensified warming of the surface mixed layer, enhancing in turn the Schmidt stability (e.g., Livingstone, 2003, Vinçon-Leite et al., 2014). Analyzing long term simulations of three lakes of different*

*average depths, Magee et al. (2017) found significant increases in summer Schmidt stability only for the two deeper study sites."*

**For part 3.3 Spatial analysis of stratification:** I mean this part is not new to me. It is very normal that the shallow part of a lake experiences less stratification compared to the deep part. I recommend that the authors also check the horizontal distribution of other thermal indices, like mixed layer depth, thermocline depth, which may provide some interesting findings.

> **Authors' response:** Thank you for these suggestions to deepen the analysis of the 3D capability of the model. We analyzed the spatial distribution of the thermocline depth. However, the definition and analysis of thermocline (or mixed layer) depth on a polymictic water body with stratification events from 1 to 10 consecutive days is quite challenging. On the long-term, no significant trend was detected, neither were horizontal patterns in its spatial distribution.
>
> To complete the study, we also extended the spatial analysis to the two ecology-derived indices (growing degree days (GDD) and number of growing days (NGD)), with the objective to detect the presence of some spatial niche favorable to phytoplankton species with high optimum temperatures (such as cyanobacteria).
>
> Three values were tested for the base temperature in the calculation of GDD and NGD (4°C, 18°C and 25°C). These values were selected since they respectively constitute baseline temperatures for (i) overall biomass growth and species able to grow at low temperature such as diatoms (after Dupuis and Hann, 2009), (ii) most phytoplankton species, normally growing at medium / high temperatures (such as dinoflagellates or green algae), and (iii) intense growth of cyanobacteria (Paerl, 2014). Results of this analysis are reported here only for GDD in Figure 2. The top three panels show the overall time-average of the annual values for GDD, calculated with the three different base temperatures (4°C, 18°C and 25°C from left to right), while bottom panels show the mean intensity of the monotonic trend for each cell in the domain, when statistically significant.
>
> Results using $T_{base}$=4°C and $T_{base}$=18°C show weak horizontal gradients (around 1% and 3%, respectively) in the distribution of GDD and no easily interpretable patterns in the intensity of their growing trend. When using $T_{base}$=25°C, horizontal gradients grow considerably, both for the overall time-average of GDD (around 4%) and for its growing trend (around 8%).
>
> While optimal thermal conditions for cold- and medium-temperature species are quite uniform in space, and have evolved quite uniformly over time, it is not the case for species with high optimum temperatures.
>
> This suggests the existence of a region, the shallower north-eastern part of the lake, particularly favorable to the development and dominance of toxic species such as cyanobacteria. Furthermore, this spatial heterogeneity is increasing over time (see fig. 2-f). This region of the lake could also become more favorable to the initiation of cyanobacteria blooms.
>
> Finally, this shows how observations taken at one single site as well as 1D models might only be partially representative of overall dynamics of a water body, especially for shallow water bodies with strong relative bathymetric variations.

[Figure]

**Figure 2:** Spatial analysis of GDD. Different base temperatures were tested for the calculation of the GDD and the relative trends; they are represented along the three columns: first column: $T_{base}=4°C$, second column $T_{base}=18°C$, third column $T_{base}=25°C$. Panels a, b and c represent, for each cell in the domain, the average over the 58 years of simulation of annual values of GDD. Panels d, e and f show, for each cell in the domain, the average interannual trend calculated through the Sen slope (all Mann-Kendall tests were statistically significant).

**For 4 Discussion:** Just as I suggested before, results in this paper is not so new compared to those from previous studies, as well as the Discussion. The innovative points, based on my opinion, is the application of 3D model to check the influence of climate change on lakes. Consequently there should be more discussion, at the last paragraph of this part, on how the heterogeneity of thermal structure in the horizontal direction can affect the aquatic ecosystems,

> **Authors' response:** The exploitation of 3-dimensional results are expanded in the revised draft, with the spatial patterns evoked in figure 2 for GDD. In the revised draft, the appropriate paragraphs in the Materials and methods, Results and Discussion sections are modified according to the results briefly discussed here for figure 2. We further develop our discussion section highlighting the new insights coming from 3D modelling on the spatial heterogeneity of the thermal dynamics in a shallow polymictic lake, based also on the spatial analysis of GDD presented in our previous response.

**Bibliographical references**

"Warm spring and summer water temperatures in small eutrophic lakes of the Canadian prairies: potential implications for phytoplankton and zooplankton". Dupuis and Hann, 2009.

"Mitigating Harmful Cyanobacterial Blooms in a Human- and Climatically-Impacted World", Paerl H. W., 2014.

"Trends and Variability of Surface Solar Radiation in Europe Based On Surface- and Satellite-Based Data Records", Pfeifroth et al., 2018.

"Analysis of Near-Surface Atmospheric Variables: Validation of the SAFRAN Analysis over France", Quintana-Seguì et al., 2008.

"Ground-based observations of clouds through both an automatic imager and human observation", Abel-Antonio Silva and Souza-Echer Mariza Pereira, 2016.

"Clearing clouds of uncertainty", Zielinka et al., 2017.

---

## Author Comment (AC2) · 21 Oct 2020

**Authors' response to referee 2**

The work belongs to the growing number of case studies on lake response to the recent climate change. Here, the authors investigated the long-term trends in a small shallow artificial lake by applying a 3-dimensional hydrodynamic model driven by a 1960-2017 regional meteorological reanalysis dataset. The combination of the study object (shallow polymictic lake) and the approach (a full 3-d model) is particularly interesting for revealing the fine mechanisms and effects of the regional climate change. The results are presented in a clear and straightforward way, but the abovementioned potential of the study is barely unfold. Except one sentence in Conclusions, the motivation for application of a 3-d lake model is not discussed, neither its advantages and disadvantages are discussed compared with simpler 1d models. It remains unclear, why should one use such a complicated model, subject to a lot of uncertainties, just to arrive at an obvious conclusion: the lake parts shallower than the mean depth of the mixed layer do not stratify. It sounds like cracking nuts with a sledgehummer. The 3d model performance is only briefly addressed. The validation was performed only on surface (mean) temperature, which is not sufficient to trust the later model results on the stratification trends. The question about the model ability to adequately reproduce vertical thermal stratification in the lake remains open.

> **Authors' response:** My coauthors and I would first like to thank you for these useful comments. In this document, we broke the first general comments into three sections in order to address the main comments sequentially and more clearly.
>
> We agree on the general remark that a stronger focus on 3D results is needed in order to highlight the novelty of the approach. This will be addressed in the revised draft both in the results and in the discussion section, as shown later on in this document. We can already say here, that 3D modelling is able to provide spatially distributed information in particular regarding areas favorable to the initiation of harmful algal blooms, or to biomass accumulation. Such results can be used to improve the monitoring design or provide stakeholders with new information for bloom control. Furthermore, in the perspective of a more generalized application of our approach, 3D modelling can benefit, even for small lakes, from the recent availability of fine resolution satellite images for instance for data assimilation (e.g. Allan et al., 2016).
>
> The concern on the validation of model performance probably comes from an unclear writing on our part. The model was indeed tested during validation at three depths (sub-surface, middle and bottom of the water column), where high-frequency measurements are available, and on two measuring sites. RMSE values were in all cases comprised between 0.96°C and 1.00°C and more specifically: Surf.:1.0°C, Mid.:0.96°C, Bott:0.96°C for site A, and Surf.:1.0°C, Mid.:0.96°C, Bott:0.99°C for site B. The validation of the model is now more clearly discussed in the revised draft through the results shown here in Figure 1, that also evaluates the timing and frequency of stable stratification events. Parity diagrams show very good performances for all layers. The comparison between modeled and simulated stable stratification events show how the model is able to correctly capture the timing of most stratification events. Some discrepancy however is generated by a threshold-induced effect in the definition of the temperature difference for stable stratification. This will allow to clearly state the overall robustness of our modeling set-up, and to discuss its limitations, when present.

[Figure]

**Figure 1:** Model performance during validation at site A. Panels a, b and c: parity diagrams between simulations and observations for the surface, middle and bottom layers, respectively. Panels d and e: visual comparison of simulated and observed water temperature at the middle layer (d) and corresponding residuals (e). Panel f: modeled (orange) vs. observed (blue) temperature difference between surface and bottom layer and relative comparison between the timing of observed and modeled stable stratification events.

A large part of discussion is dedicated to the effect of climate change on the transient stratification development in shallow polymictic lakes. However, the stratification indices used in the analysis—Schmidt stability and the total stratification duration—are rather relevant to oligomictic (di- and monomictic) lakes. Neither duration of the longest stratification period, nor the frequency of stratification events are analyzed.

> **Authors' response:** The Schmidt stability and SSD can give relevant information also in the case of polymictic lakes, and both indices are used in various studies on shallow polymictic water bodies (e.g. Magee et al. 2017, Moras et al. 2019).
> We agree that an analysis of the frequency and length of stratification events is a very interesting complement to the two previously mentioned indices for a polymictic lake. The results of this analysis showed that no significant trend can be found over the long term for the annual number of stable stratification events. In contrast, a shift in the average duration of stratification events can be detected. This is shown in Figure 2, where the double cumulative curve of annual SSD and annual number of stable stratification events is plotted.

A change point analysis of the underlying linear trend shows a break point between 1988 and 1989 with a shift in the average duration of a stable stratification event from 3 to 4 days. This can be expected to influence phytoplankton growth and is discussed in the revised manuscript.

[Figure]

**Figure 2:** Double cumulative curves for the annual number of SSD (x axis) and the annual number of stable stratification events (y axis).

The indices used for the warming effect on the net biological production are also questionable: temerature as a measure of the growing season is weakly justified in lake ecosystems, especially for the climate under consideration. The "number of growing days" (NGD) in the authors' formulation is always clearly above 300, so the whole 365-days long year cycle can be a priori assumed as production-favorable in terms of temperature. Trends in GDD and NGD do not seem to be representative for any biotic processes. In particular, because high temperatures can work as a stress factor inhibiting both primary production and growth rates of higher organisms. In general, Discussion presents a lot of common knowledge but is weakly connected to the results from the study.

In summary: the study uses a promising approach and a solid dataset, but, in its current form, presents little advance on the subject under investigation. A stronger focus on the abilities of 3-d modeling for climatic lake studies and intermittent stratification dynamics of shallow polymictic lakes would strengthen this otherwise well-designed and clearly structured study.

**Authors' response:** Concerning the temperature based indices for aquatic environments, their application is indeed quite rare but recently growing (e.g. Dupuis and Hann, 2009, Rlaston et al., 2014, Sterner et al., 2020). In order to link the GDD and NGD not only to the overall growth of biomass but also to the growth of specific algal groups with higher optimum temperatures, two other baseline temperatures have been introduced in our analysis.

Three values were therefore tested for the base temperature in the calculation of GDD and NGD (4°C, 18°C and 25°C). These values were selected since they respectively constitute baseline temperatures for (i) overall biomass growth and species able to grow at low temperature such as diatoms (after Dupuis and Hann, 2009), (ii) most phytoplankton species, normally growing at medium / high temperatures (such as dinoflagellates or green algae), and (iii) intense growth of cyanobacteria (Paerl, 2014). These results have been analyzed over time and space and will be included in the revised draft. Such results are shown here in figure 3 for GDD, where the top 3 panels show the overall time-average of the annual values for GDD, calculated with the three different base temperatures (4°C, 18°C and 25°C from left to right), while bottom panels show the mean intensity of the monotonic trend for each cell in the domain, when statistically significant.

For $T_{base}=4°C$ and $T_{base}=18°C$ weak horizontal gradients (around 1% and 3%, respectively) can be found in the spatial distribution of time averaged GDD. No easily interpretable patterns can be seen in the intensity of their

growing trend. Horizontal gradients grow considerably when using $T_{base}$=25°C, both for the overall time-average of GDD (around 4%) and for its growing trend (around 8%).

While optimal thermal conditions for cold- and medium-temperature species are quite uniform in space, and have evolved quite uniformly over time, it is not the case for species with high optimum temperatures.

This suggests the existence of a region, the shallower north-eastern part of the lake, particularly favorable to the development and dominance of toxic species such as cyanobacteria. Furthermore, this spatial heterogeneity is increasing over time (see fig. 3-f). This region of the lake could also become more favorable to the initiation of cyanobacteria blooms.

Finally, this shows how observations taken at one single site as well as 1D models might only be partially representative of overall dynamics of a water body, especially for shallow water bodies with strong relative bathymetric variations.

In order to take this analysis into account, in the revised draft the relative parts in the Materials and Methods, Results and Discussion sections are appropriately modified. The results and the limitations of this approach for linking hydrodynamics and ecology (as water temperature is a key factor but not the only one influencing primary production) are further discussed in the revised draft.

[Figure]

**Figure 3:** Spatial analysis of GDD. Different base temperatures were tested for the calculation of the GDD and the relative trends; they are represented along the three columns: first column: $T_{base}$=4°C, second column $T_{base}$=18°C, third column $T_{base}$=25°C. Panels a, b and c represent, for each cell in the domain, the average over the 58 years of simulation of annual values of GDD. Panels d, e and f show, for each cell in the domain, the average interannual trend calculated through the Sen slope (all Mann-Kendall tests were statistically significant).

**Here are some specific remarks:**

- The model uses constant cloudiness as input, which is quite strange, especialy, taking into accout significant long-term trends in solar radiation. Why the real variability of cloud cover (or long-wave atmospheric radiation) was not used? Can you estimate the resulting errors in the model output?

> **Authors' response:** We agree that it would be preferable to use cloudiness measurements. Cloud cover measurements were available from the closest meteorological station, 24 km away, but not for the 8x8 km SAFRAN reanalysis cell, contrary to other meteorological variables.
>
> The model was tested during calibration and validation using two different modules for heat exchange, the Ocean model with time series of cloud cover, and the Murakami model which is implemented with a constant cloud cover in Delft3D. Overall, water temperature was slightly better modeled by the Murakami model with a specifically calibrated value for cloud cover. For instance, for the year 2016, RMSE at site A was around 1.1°C for all three depths with the Ocean heat exchange model, while it was around 0.8°C for the Murakami heat exchange model.
>
> This can be explained on the first hand by the high uncertainty that affects ground-based cloud cover observations (Silva and Souza-Echer, 2015, Zelinka et al., 2017) and by some gaps found in the series, and on the second hand by the different source of these data from the other meteorological input. Furthermore, a preliminary analysis that we carried out on 19 years (from 2000 to 2018) of cloud cover data from the closest meteorological station, showed an overall very low seasonality in the series, (with 7 oktas (87%) being the most probable value for all seasons and 65% being the overall mean value) and no interannual trend. The absence of strong climatic trends for the area of interest in terms of cloud cover is confirmed by Pfeifroth et al. (2018).
>
> We therefore decided to use the constant calibrated value of sky cloudiness in order to achieve the best results in terms of simulated water temperature against our set of hourly averaged water temperature observations. In any case, this led to very good model performances during both calibration and validation.
>
> Finally, we think that the calibration of the parameter to a value of 80% does not introduce considerable biases in our work, even in the long term study, also given the non-negligible uncertainty that affects cloud cover observations.

- Also, a constant water transparency is used in the long-term model runs, despite the data indicate a strong transparency variability on seasonal scales. How this assumption affected the model predictions on stratification patterns? Will the time-variable Secchi depth change the modeling results?

> **Authors' response:** We chose a constant and calibrated value of the Secchi depth for two main reasons. First of all, no Secchi depth records were available for the long term simulation (before Spring 2015). Furthermore, the Secchi depth in the study site can vary suddenly between 0.8 m and the whole water column depth according to short term (one or two weeks) phytoplankton bloom events that occur between February and October. Because of the lack of long term data and the difficulty in defining a clear seasonal pattern for Secchi depth, we decided to test if a constant value would be sufficient to get good performances. We therefore performed a calibration and then used the calibrated value as a constant input of the model. As we obtained very good performances during both calibration and validation, we decided to keep it constant for this study. However, it is indeed a strong simplification that is further discussed in the revised draft.

- I do not believe that the trends in solar radiation and wind are monotonic (Fig. 3bc). A change point detection analysis should be performed here (e.g. B.K. and Tsay, R.S., 2002. Bayesian methods for change-point detection in long-range dependent processes. Journal of Time Series Analysis, 23(6), pp.687-705., or any other similar approach) with subsequent piecewise trend estimation.

> **Authors' response:** Thank you for this remark. Just like you, we had thought of applying piecewise trend estimation on this two variables, but as the overall monotonic trends were significant, we did not do it in the initial version of the article. To follow your suggestion, we carried out a change point analysis using the Matlab function "findchangepts" (Matlab 2020b). It showed for both solar radiation and wind speed the existence of a change point at the end of the 1980s (1988 and 1987, respectively), as shown here in Figure 4. This fact was only qualitatively addressed in the submitted version, but is more clearly discussed in the revised draft.

[Figure]

**Figure 4:** Mean annual averages for air temperature, solar radiation and wind speed, with change point detection.

**Bibliographical references**

"Spatial heterogeneity in geothermally-influenced lakes derived fromatmospherically corrected Landsat thermal imagery andthree-dimensional hydrodynamic modelling". Allan et al., 2016.

"Warm spring and summer water temperatures in small eutrophic lakes of the Canadian prairies: potential implications for phytoplankton and zooplankton". Dupuis and Hann, 2009.

"Response of water temperatures and stratification to changing climate in three lakes with different morphometry". Magee et al., 2017.

"Historical modelling of changes in Lake Erken thermal conditions". Moras et al., 2019.

"Mitigating Harmful Cyanobacterial Blooms in a Human- and Climatically-Impacted World", Paerl H. W., 2014.

"Analysis of Near-Surface Atmospheric Variables: Validation of the SAFRAN Analysis over France", Quintana-Seguì et al., 2008.

"Trends and Variability of Surface Solar Radiation in Europe Based On Surface- and Satellite-Based Data Records", Pfeifroth et al., 2018.

"Temperature dependence of an estuarine harmful algal bloom: Resolving interannual variability in bloom dynamics using a degree day approach". Ralston et al., 2014.

"Ground-based observations of clouds through both an automatic imager and human observation", Abel-Antonio Silva and Souza-Echer Mariza Pereira, 2016.

"A first assessment of cyanobacterial blooms in oligotrophic Lake Superior". Sterner et al., 2020.

"Clearing clouds of uncertainty", Zielinka et al., 2017.

---

## Author Response (AR1)

**The thermal response of small and shallow lakes to climate change: new insights from 3D hindcast modelling.**

Francesco Piccioni[1], Céline Casenave[2], Bruno Jacques Lemaire[1,3], Patrick Le Moigne[4], Philippe Dubois[1], and Brigitte Vinçon-Leite[1]

[1]LEESU, Ecole des Ponts ParisTech, Univ Paris Est Créteil, Marne-la-Vallée, France
[2]MISTEA, Université Montpellier, INRAE, Institut Agro, Montpellier, France
[3]AgroParisTech, Paris, France
[4]CNRM, Université de Toulouse, Météo-France, CNRS, Toulouse, France

**Correspondence:** Francesco Piccioni (francesco.piccioni@enpc.fr)

**Author's response**

First of all, my co-authors and I would like to thank the Reviewers and the handling Editor for the time and interest taken in evaluating our work, as well as for their suggestions that helped improve our analysis.

5  In the present work, the thermal regime of a shallow urban lake is reconstituted over six decades (between 1960 and 2017) with a 3D thermal-hydrodynamic model, after calibration and validation. Simulation results are analyzed over time (for long-term monotonic trends), and space (for spatial heterogeneity), through a series of indices that characterize the stratification dynamics and highlight the relation between temperature and primary production. The work was initially submitted on the 10th July 2020, and has already undergone a first round of reviews.

10

As stated by the Editor in his final report, both reviews were constructive and quite similar in their major concerns, to which we have already addressed a first response (available at the interactive discussion page: https://esd.copernicus.org/preprints/esd-2020-51/#discussion). Both reviewers asked some clarifications on the choice of constant values for some of the parameters used in our modelling design (in particular the Secchi depth and sky cloudiness). However, no substantial remarks were
15  made on the technical robustness of our working design.

The major concern regarded the novelty of this study, as the most distinctive element of our work, the 3D modelling approach, was not adequately exploited. Furthermore, doubts were raised on the biological relevance of the two indices chosen here as proxies for biomass production (namely, the growing degree days (GDD) and the number of growing days (NGD)). In particular, the choice of the temperature threshold for the calculation of the two indices was questioned as well as the lack of an upper
20  limit for growth taking into account high-temperature stress.

According to the main remarks pointed out in the reviews and summarized above, we substantially modified the analysis

of model results, with the objective of better highlighting in the revised version the potential of a 3D modelling approach and its relevance even for small water bodies. Namely, the main changes we have done concern the exploitation of the 3D results and the definition of the indices representing primary production. In particular, compared to what was initially suggested in our first reply to the reviewers, the analysis of the 3D results was expanded and has surpassed the ideas initially proposed in the reply.

In this revised draft we propose a more structured study of the spatial heterogeneity that focuses on the distribution of both stratification and potential primary production. This was done, after the Editor's and reviewers suggestions, by plotting the probability density function of some indices, which allows to visualize the evolution over time of the spatial distribution of these indices. The proxies for primary production were also changed, according to the remarks of the reviewers. An upper limit has been added in the expression of the growing degree days (GDD) to account for a possible stress induced by high temperature. The number of growing degree days (NGD) is no more considered in the revised version of the paper. It has been replaced by an other index that is the mean annual value of the thermal growth rate of cyanobacteria.

The major changes made to the original draft are described hereby in a detailed point-to-point list. In the following paragraphs, italics is used for extracts of the initially submitted version, while the blue colour identifies the parts of the text coming from the revised manuscript.

**Indices for primary production**

In the first version of our manuscript, two indices were used to characterize primary production from a thermal standpoint: the growing degree days (GDD) and the number of growing days (NGD). The GDD were defined as follows:

*After Dupuis and Hann (2009), GDD were calculated as follows:*

$$GDD(t) = \sum_{i=t_0}^{t} max\left\{0, (T_i - T_{base}) \cdot \Delta t\right\} \tag{1}$$

*where $t$ is the time (here in days) with $t_0$ the reference day to start the calculation, $\Delta t$ is the time step (equal to 1 day), $T_i$ is the daily average of the modeled surface water temperature of day $i$ and $T_{base}$ is a physiological threshold below which growth does not occur. [...] As in Dupuis and Hann (2009), the value of 4°C was selected for $T_{base}$. Such value was chosen to be a representative baseline for the growth of the whole phytoplankton community in Lake Champs-sur-Marne, generally composed of cyanobacteria, green algae and diatoms.*

While for the NGD:

*The number of growing days (NGD) at day $t$ is defined as the number of days during which $(T_i - T_{base}) > 0$ from day $i = t_0$ to $i = t$. It represents a proxy for the duration of the period favorable to growth for the phytoplankton community.*

The main remarks done by the Reviewers regarded the choice of such a low value for the base temperature and in general the relevance of the NGD in relation to biomass production. Furthermore, also the lack of an upper limit for the GDD that takes into account reduced phytoplankton growth under high temperature was pointed out.

60     According to these remarks, in the revised version the indices related to primary production were changed. We focus our attention only on cyanobacteria, and not on the whole phytoplankton community present in the study site. The NGD are completely abandoned, while an upper limit for cyanobacteria growth is added for the GDD. Furthermore, the concept of thermal growth rate (GR) is introduced. This allows to thoroughly address the effect of water temperature on potential cyanobacteria production, while taking into account growth reduction at low or extremely high temperatures.

65 The definition of the new indices (GR and GDD) is now as follows:

Under the assumption of nutrient and light availability, phytoplankton growth rate can be modelled, for different species, as a function of temperature as follows (Bernard and Rémond, 2012) :

70   $$k(T) = k_{opt} \frac{(T - T_{max})(T - T_{min})^2}{(T_{opt} - T_{min})[(T_{opt} - T_{min})(T - T_{opt}) - (T_{opt} - T_{max}(T_{opt} + T_{min} - 2T))]}, \ \forall \, T \in [T_{min}, T_{max}] \tag{2}$$

where $k_{opt}$ is the optimal growth rate, $T_{min}$ the minimal temperature, $T_{opt}$ the optimal temperature and $T_{max}$ the maximal temperature. The model parameters were calibrated by You et al. (2018) through experimental data to describe the response to water temperature of *Microcystis aeruginosa*, a species of cyanobacteria present in Lake Champs-sur-Marne and often

75 dominant in freshwater bodies globally. The same values are used in this work:

$$k_{opt} = 0.74 \text{d}^{-1}, \ T_{min} = 0°C, \ T_{opt} = 27.5°C, \ T_{max} = 38.4°C. \tag{3}$$

*Microcystis aeruginosa* is thought to be favored by the warmer water temperatures induced by climate change. However, the curve obtained from eq. 2 and 3 (shown in figure 1), is here more generally intended to be representative of the typical thermal response of cyanobacteria with high optimum temperature. Mean annual and seasonal (according to 2.3) growth rates

80 are calculated through eq. 2 using simulated surface water temperature, and analysed over space and time.

    The growing degree days are a weather based indicator for biological growth, widely used in the field of agronomy. Based on air temperature, it gives an estimate of the rate of development and of the span of the growing season for terrestrial plants and insects. It is a useful indicator capable to link global warming and biology (Grigorieva et al., 2010; Schlenker et al., 2007). Approaches based on GDD have been increasingly applied to phytoplankton communities and fisheries (e.g. Gillooly, 2000;

85 Neuheimer and Taggart, 2007; Ralston et al., 2014; Dupuis and Hann, 2009), in order to correlate water temperature and phytoplankton growth while taking into account interannual variability. After Dupuis and Hann (2009), GDD were calculated as follows:

[Figure]

**Figure 1.** Thermal growth rate calculated after equation 2. The horizontal dashed line for GR=0.2 d$^{-1}$ meets the curve at the temperature limits for the calculation of the GDD (10°c and 37°C, respectively).

$$GDD(t) = \sum_{i=t_0}^{t} a_i \cdot (T_i - T_{base}) \cdot \Delta t, \quad \text{with } a_i = \begin{cases} 1 & \text{if } T_{base} < T_i < T_{sup} \\ 0 & \text{elsewhere} \end{cases} \tag{4}$$

90

where $t$ is the time (here in days) with $t_0$ the reference day to start the calculation, $\Delta t$ is the time step (equal to 1 day), T$_i$ is the daily average of the modeled surface water temperature of day $i$ and T$_{base}$ (respectively $T_{sup}$) is a physiological threshold below which (respectively above which) growth does not occur. Compared to the formulation found in Dupuis and Hann (2009), an upper limit for growth was introduced here (T$_{sup}$) to take into account high temperature stress. Our focus here is, as

95    for the GR, on cyanobacteria. After Thomas et al. (2016) and based on the latitude of the study site, we set the base temperature at 10°C and the upper limit for growth at 37°C. This results in considering, for the cacluation of the GDD, only temperatures that yield to a GR above 0.2 d$^{-1}$ (see figure 1).

GDD can be calculated on an annual or a seasonal basis by adjusting the values of $t_0$ and $t$. Annual GDD are calculated from the first of January until the 31st of December. Seasonal GDD are obtained according to the definitions of section 2.3.

100    **3D analysis**

In the revised version, the spatial analysis is not limited to the stratification dynamics but encompasses also the distribution of surface water temperature, thermal growth rate and growing degree days. Furthermore, a more structured study of the spatial heterogeneity is proposed. Its main focus is on the distribution of stratification and surface water temperature and on their impact on potential cyanobacteria production. The horizontal distribution of the variables under examination is fitted to a non

105    parametric probability density function (the Kernel function) and finally plotted as a heat map.

Following is its description as in the revised version. The related paragraphs in the results and discussion sections were changed

according to this new methodology and can be found in the revised manuscript.

The long-term evolution and the spatial variability of the thermal regime of Lake Champs-sur-Marne was further analysed exploiting the three-dimensional model simulations. Mean annual surface water temperature, annual SSD, mean annual GR and annual GDD were computed on the whole computational domain, with the objective of investigating the relation between climate change and time evolution of the spatial distribution of these variables. For each variable $x$, the overall mean annual value $x_m$ (averaged over the complete domain) and the deviation from the mean value $(x - x_m)$ have then been computed. In order to quantify the spatial heterogeneity of these variables, the probability distribution of the deviation from the mean value of each variable was finally calculated on the computational domain and fitted, for each year, with a non-parametric Kernel probability distribution through the Matlab *pdf* function. The resulting probability density function (PDF) was plotted over time as a heat map and the mean value as a simple line plot. This allows to visualize both the time and the spatial evolution of the variable under consideration, by looking at the mean value and at the range of values characterized by a non-zero probability.

During stably stratified periods, cyanobacteria are favored over other algal groups because of their ability to move within the water column and possibly float towards the water surface (Humphries and Lyne, 1988; Wagner and Adrian, 2009; You et al., 2018). For this reason, the spatial analysis of the GR and GDD was completed, by calculating these two indices only on stable stratified days during each year. The obtained GR were further averaged for each cell over the local number of stably stratified days. Cells that showed an annual number of SSD<10 where discarded from this analysis. Finally, the resulting modified indices were analysed over space and time as described above by using a non-parametric Kernel probability distribution as an approximation of the PDF for each simulated year.

**Additional modifications**

According to the suggestions found in the reviews other additional modifications were done to our text. They didn't entail major changes to the text, but are listed hereafter.

– The process and the results of validation were more clearly explained, especially through additional figures showing more in detail the model performances in terms of water temperature (parity diagrams between simulation and observations were added) and stratification dynamics (with a visual comparison of the observed and simulated stratification events).

– In order to further characterize the dynamics of the stratification in a polymictic water body, besides the calculation of the stably stratified days (SSD) and the Schmidt stability index, also the annual frequency of stable stratification events and the duration of the longest consecutive stratification event were analyzed.

– The title of the article was slightly changed from "The response of small and shallow lakes to climate change: new insights from hindcast modelling", to: "The thermal response of small and shallow lakes to climate change: new insights from 3D hindcast modelling".

**References**

Bernard, O. and Rémond, B.: Validation of a simple model accounting for light and temperature effect on microalgal growth, Bioresource

140    Technology, 123, 520–527, https://doi.org/10.1016/j.biortech.2012.07.022, 2012.

Dupuis, A. P. and Hann, B. J.: Warm spring and summer water temperatures in small eutrophic lakes of the Canadian prairies: potential implications for phytoplankton and zooplankton, J Plankton Res, 31, 489–502, https://doi.org/10.1093/plankt/fbp001, 2009.

Gillooly, J.: Effect of body size and temperature on generation time in zooplankton, Journal of Plankton Research, 22, https://doi.org/10.1093/plankt/22.2.241, 2000.

145 Grigorieva, E., Matzarakis, A., and De Freitas, C.: Analysis of growing degree-days as climate impact indicator in a region with extreme annual air temperature amplitude, Climate Research, 42, 143–154, https://doi.org/10.3354/cr00888, 2010.

Humphries, S. E. and Lyne, V. D.: Cyanophyte blooms: The role of cell buoyancy1, Limnology and Oceanography, 33, 79–91, https://doi.org/https://doi.org/10.4319/lo.1988.33.1.0079, _eprint: https://aslopubs.onlinelibrary.wiley.com/doi/pdf/10.4319/lo.1988.33.1.0079, 1988.

150 Neuheimer, A. B. and Taggart, C. T.: The growing degree-day and fish size-at-age: The overlooked metric, Canadian Journal of Fisheries and Aquatic Sciences, 64, 375–385, 2007.

Ralston, D. K., Keafer, B. A., Brosnahan, M. L., and Anderson, D. M.: Temperature dependence of an estuarine harmful algal bloom: Resolving interannual variability in bloom dynamics using a degree day approach, Limnol. Oceanogr., 59, 1112–1126, https://doi.org/10.4319/lo.2014.59.4.1112, 2014.

155 Schlenker, W., Hanemann, W. M., and Fisher, A. C.: Water Availability, Degree Days, and the Potential Impact of Climate Change on Irrigated Agriculture in California, Climatic Change, 81, 19–38, https://doi.org/10.1007/s10584-005-9008-z, 2007.

Thomas, M. K., Kremer, C. T., and Litchman, E.: Environment and evolutionary history determine the global biogeography of phytoplankton temperature traits, Global Ecology and Biogeography, 25, 75–86, https://doi.org/https://doi.org/10.1111/geb.12387, _eprint: https://onlinelibrary.wiley.com/doi/pdf/10.1111/geb.12387, 2016.

160 Wagner, C. and Adrian, R.: Cyanobacteria dominance: Quantifying the effects of climate change, Limnology and Oceanography, 54, 2460–2468, https://doi.org/10.4319/lo.2009.54.6_part_2.2460, 2009.

You, J., Mallery, K., Hong, J., and Hondzo, M.: Temperature effects on growth and buoyancy of Microcystis aeruginosa, J Plankton Res, 40, 16–28, https://doi.org/10.1093/plankt/fbx059, publisher: Oxford Academic, 2018.

---

## Editor Decision (ED1)

Decision to sd-2020-51

Submitted on 10 Jul 2020
The response of small and shallow lakes to climate change: new insights from hindcast modelling

Francesco Piccioni, Céline Casenave, Bruno Jacques Lemaire, Patrick Le Moigne, Philippe Dubois, and Brigitte Vinçon-Leite

Dear Dr. Piccioni,

I have now looked at the MS and your responses to the reviews and want to communicate my decision to you as Handling Editor.

Both reviews had been constructive and also quite similar in their major concerns. The major problem from my point of view is the limited novelty of your work as the literature is already flooded with simulation studies on climate change effects on lakes, at least on the case study level. I take this concern very serious as ESD is seeking novel research contents!

Although your MS is well written and technically sound, I am still not satisfied with the current version of the MS and your responses to the reviewers. But I also think that the work has potential and may become a contribution after a serious revision.

A strong point is the 3D model approach, which is (still) not yet well exploited. Besides working with specific thresholds and indices, you could also analyse and display probability density functions of your variables. For example, how is the pdf of surface water temperatures (during summer or winter or whole year?) change from decade to decade (it does not make sense to do that for each year separately). The same approach can be used to analyse spatial heterogeneity, i.e. show horizontal variability by calculating the temperature anomalies (difference between horizontally averaged temperature and single-cell temperatures) over space and then show pdfs or any other clever form of data aggregation. The pdf will also enable a more detailed statistical approach and exploits the 3D model setting.

I also believe the concepts of GDD and NGD are weak concepts as they miss important aspects (e.g. heat stress as mentioned by one reviewer). For example, you could similarly evaluate the number of days with water temperature above 25°C in order to characterise heat waves or heat stress. At best, the critical temperature is not just 25°C ( I chose arbitrarily), but is rather a temperature that has a reliable ecological interpretation, e.g. the temperature from which cyanophytes may become dominant. I remember that Jeff Huismann has worked on that, but I do not want to guide you through the literature. Instead, you should make a thorough literature review and come up with a (or several) meaningful criterion.

I encourage you to invest more energy in the evaluation of your simulation results as the simulation itself and the corresponding modelling task has not much novelty! I expect that you not simply my following my proposed ideas but also develop own ways.

If you decide to revise, a new evaluation will be required and I cannot guarantee acceptance of the MS.

Best whishes,

Karsten Rinke (Handling Editor)